# Demystifying Spectral Feature Learning for Instrumental Variable Regression

**Dimitri Meunier**[*]
Gatsby Unit, UCL

**Antoine Moulin**
Universitat Pompeu Fabra

**Jakub Wornbard**
Gatsby Unit, UCL

**Vladimir R. Kostic**
Istituto Italiano di Tecnologia & University of Novi Sad

**Arthur Gretton**
Gatsby Unit, UCL

## Abstract

We address the problem of causal effect estimation in the presence of hidden confounders, using nonparametric instrumental variable (IV) regression. A leading strategy employs ***spectral features*** - that is, learned features spanning the top eigensubspaces of the operator linking treatments to instruments. We derive a generalization error bound for a two-stage least squares estimator based on spectral features, and gain insights into the method's performance and failure modes. We show that performance depends on two key factors, leading to a clear taxonomy of outcomes. In a ***good*** scenario, the approach is optimal. This occurs with strong ***spectral alignment***, meaning the structural function is well-represented by the top eigenfunctions of the conditional operator, coupled with this operator's slow eigenvalue decay, indicating a strong instrument. Performance degrades in a ***bad*** scenario: spectral alignment remains strong, but rapid eigenvalue decay (indicating a weaker instrument) demands significantly more samples for effective feature learning. Finally, in the ***ugly*** scenario, weak spectral alignment causes the method to fail, regardless of the eigenvalues' characteristics. Our synthetic experiments empirically validate this taxonomy. We further introduce a practical procedure to estimate these spectral properties from data, allowing practitioners to diagnose which regime a given problem falls into. We apply this method to the dSprites dataset, demonstrating its utility.

## 1 Introduction

We study the nonparametric instrumental variable (NPIV) model [1, 32]

$$Y = h_0(X) + U, \quad \mathbb{E}[U \mid Z] = 0, \quad Z \not\!\perp X, \tag{1}$$

where $X$ is the treatment, $Y$ the outcome, $Z$ the instrument, $U$ an unobserved confounder, and $h_0 \in L_2(X)$ is the structural function to be learned, where we denoted $L_2(X)$ the $L_2$ space corresponding to the distribution $\mathbb{P}_X$. We assume no common confounder of $Z$ and $Y$. To illustrate this setting, consider a simplified example inspired by [20]. Suppose we want to determine how ticket price ($X$) affects the number of flight tickets sold ($Y$). A potential confounder ($U$) could be a major event (*e.g.*, a large conference or festival) occurring at the destination. This event would likely increase demand for flights, and airline pricing algorithms would result in more expensive seats being sold; the confounder $U$ influences both $X$ and $Y$. A naive regression of sales on price might then misleadingly suggest that higher prices lead to higher sales. This occurs because the analysis fails to account for the event, which independently drives up both demand (sales) and prices. To obtain a less biased estimate of the price effect, we could use an instrument ($Z$), such as the cost of jet fuel. The

---

[*]direct correspondence to dimitri.meunier.21@ucl.ac.uk

cost of fuel is a plausible instrument because: (i) it likely affects ticket prices ($X$) as airlines adjust fares based on operational costs, and (ii) it is unlikely to be directly correlated with whether a major event ($U$) is happening at the destination, thus satisfying $\mathbb{E}[U \mid Z] = 0$. By using fuel cost as an instrument, we can isolate the variation in ticket prices that is independent of the unobserved demand shock caused by the event, allowing for a more accurate estimation of the structural function $h_0$.

Central to the NPIV model is the ***conditional expectation operator*** $\mathcal{T} : L_2(X) \to L_2(Z)$, defined by $\mathcal{T}h(Z) = \mathbb{E}[h(X) \mid Z]$. Estimating $h_0$ amounts to inverting this operator from the following integral equation

$$\mathcal{T}h_0 = r_0, \quad \text{where } r_0(Z) \doteq \mathbb{E}[Y \mid Z]. \tag{2}$$

In practice, the operator $\mathcal{T}$ is unknown and must be estimated from data. Moreover, even if $\mathcal{T}$ were known, its inverse is typically unbounded, making recovery of $h_0$ an ill-posed inverse problem that requires regularization.

A classical approach to solving NPIV is nonlinear two-stage least squares (2SLS), which projects $X$ and $Z$ into suitable feature spaces and performs regression in these representations. Let $\varphi$ and $\psi$ denote feature maps for $X$ and $Z$, respectively. In **Stage 1**, one estimates the conditional expectation $\mathbb{E}[\varphi(X) \mid Z]$ by regressing $\varphi(X)$ on $\psi(Z)$, yielding an approximation of the form $\mathbb{E}[\varphi(X) \mid Z] \approx A\psi(Z)$, for some linear map $A$. This step isolates the component of $\varphi(X)$ that is predictable from $Z$, filtering out variation due to the unobserved confounder. In **Stage 2**, the outcome $Y$ is regressed on the predicted features $\mathbb{E}[\varphi(X) \mid Z]$, yielding an estimator of the structural function $h_0$, which is modeled as a linear combination $h_0(X) \approx \beta^\top \varphi(X)$, for some coefficient vector $\beta$. Given that the feature maps $\varphi$ and $\psi$ can be fixed (*e.g.*, polynomials, splines, or radial basis functions from a kernel) or learned from data (*e.g.*, using neural networks), a natural question arises: ***What specific information must these features encode to enable efficient estimation?***

The algorithm recently proposed by [36], which learns features by minimizing a spectral contrastive loss, suggests that the answer lies in the spectral structure of $\mathcal{T}$. In this paper, we rigorously investigate this perspective. We show that the performance of this approach hinges on two key factors: the ***spectral alignment*** of the structural function $h_0$ with the operator $\mathcal{T}$, and the rate of its ***singular value decay***. Namely, we identify three distinct regimes governing the effectiveness of spectral methods for causal inference:[2]

1. The ***good*** regime, where the structural function aligns strongly with the leading eigenspaces of the conditional expectation operator $\mathcal{T}$, and the singular values of $\mathcal{T}$ decay slowly;

2. The ***bad*** regime, where spectral alignment holds, but rapid singular value decay leads to instability and degraded estimation quality;

3. The ***ugly*** regime, where the structural function is misaligned with the top eigenspaces of $\mathcal{T}$, rendering spectral methods ineffective regardless of decay rate.

**Overview of contributions.** We investigate the effectiveness of spectral features in NPIV estimation, grounding their empirical success in a principled theoretical analysis. Specifically:

• By leveraging an upper bound on the generalization error for the 2SLS estimator and specializing it to spectral features, we characterize the conditions under which each regime arises.

• We clarify the theoretical foundation of the state-of-the-art causal estimation algorithm of [36] by showing that it implements a 2SLS estimator with learned features that explicitly target the leading eigenspaces of the conditional expectation operator $\mathcal{T}$. Although [36] suggested a heuristic link to the Singular Value Decomposition (SVD) of $\mathcal{T}$, we make this connection explicit by proving that their contrastive objective corresponds to a Hilbert-Schmidt approximation of $\mathcal{T}$.

• We empirically validate the "good–bad–ugly" taxonomy through synthetic experiments designed to isolate the effects of spectral alignment and singular value decay on generalization error. We further introduce a practical, data-driven procedure to estimate these key spectral properties (spectral alignment and singular value decay), allowing practitioners to diagnose their problem's regime. We demonstrate this procedure on the dSprites dataset [30].

---

[2]The "good–bad–ugly" terminology is borrowed from [24], who used it to describe performance regimes in Koopman operator learning. We adopt the same naming convention in a different context: spectral feature learning for causal inference. The phrase itself, of course, originates from the classic [27].

**Related work.** Instrumental variable regression is often implemented as a two-stage procedure, where the key design choice is the feature representation. Fixed features include polynomials, splines, or kernel methods [3, 6, 7, 31, 35], while learned features are either derived from the conditional expectation operator $\mathcal{T}$ [36] or jointly optimized with the 2SLS objective [23, 33, 40]. Some approaches replace first-stage regression with conditional density estimation [11, 18, 20, 28, 34].

An alternative line of work reformulates IV regression as a saddle-point optimization problem to avoid the bias introduced by nested conditional expectation estimation [2, 13, 29, 38, 43]. These methods frame estimation as a min-max game between a candidate solution $h$ and a dual witness function enforcing the IV moment condition. This approach sidesteps explicit conditional expectation estimation but relies on selecting tractable, fixed function classes to ensure the well-posedness of the optimization problem.

Features can be learned in various ways. One line of work learns them jointly with the regression objective, as in [16, 23, 33, 40], who propose end-to-end frameworks that simultaneously optimize over the feature representations and second-stage parameters. In contrast, [36, 38] propose to learn features that reflect the eigenstructure of $\mathcal{T}$, independently of the outcome $Y$, using a spectral contrastive loss. This may appear suboptimal, since incorporating $Y$ could, in principle, yield more informative representations, however it avoids the optimization difficulties inherent to nonconvex, joint objectives involving all three variables $(X, Y, Z)$. This decoupled spectral approach was shown in certain experimental settings to outperform end-to-end alternatives, as demonstrated in [36]. Our work places this empirical finding in context, since it likely arises due to the experiments in that work satisfying the "good" regime in the present work. We provide further discussion in Appendix D.

**Structure of the paper.** Section 2 introduces the notation and preliminaries. Section 3 reviews the Sieve 2SLS estimator and presents generalization bounds highlighting the role of spectral features. Section 4 shows how to learn such features via a contrastive loss and connects this to the method of [36]. Section 5 presents a practical procedure to estimate the spectral properties from data, enabling an empirical diagnosis of our taxonomy. Section 6 provides synthetic experiments, including an application to the dSprites dataset, that validate our theoretical taxonomy. Proofs are deferred to the Appendix.

## 2 Preliminaries

### 2.1 Background

**Probability Spaces and Function Spaces.** $Y$ is defined on $\mathbb{R}$, while $X$ and $Z$ take values in measurable spaces $\mathcal{X}$ and $\mathcal{Z}$, respectively, endowed with their $\sigma$-fields $\mathcal{F}_{\mathcal{X}}$ and $\mathcal{F}_{\mathcal{Z}}$. We let $(\Omega, \mathcal{F}, \mathbb{P})$ be the underlying probability space with expectation operator $\mathbb{E}$. For $R, S \in \{X, Z, (X, Z)\}$, let $\pi_R$ denote the push-forward of $\mathbb{P}$ under $R$, and let $\pi_R \otimes \pi_S$ denote the product measure on the corresponding product $\sigma$-field, defined by $(\pi_R \otimes \pi_S)(A \times B) = \pi_R(A)\pi_S(B)$ for measurable sets $A, B$. For $R \in \{X, Z\}$ and corresponding domain $\mathcal{R} \in \{\mathcal{X}, \mathcal{Z}\}$, we abbreviate $L_2(R) \equiv L_2(\mathcal{R}, \pi_R)$ as the space of measurable functions $f : \mathcal{R} \to \mathbb{R}$ such that $\int |f|^2 \, d\pi_R < \infty$, defined up to $\pi_R$-almost everywhere equivalence.

**Operators on Hilbert Spaces.** Let $H$ be a Hilbert space. For a bounded linear operator $A$ acting on $H$, we denote its operator norm by $\|A\|_{\mathrm{op}}$, its Hilbert–Schmidt norm by $\|A\|_{\mathrm{HS}}$, its Moore–Penrose inverse by $A^{\dagger}$, and its adjoint by $A^{\star}$. For finite-dimensional operators, the Hilbert–Schmidt norm coincides with the Frobenius norm. We denote the range and null space of $A$ by $\mathcal{R}(A)$ and $\mathcal{N}(A)$, respectively. Given a closed subspace $M \subseteq H$, we write $M^{\perp}$ for its orthogonal complement, $\overline{M}$ for its closure, and $\Pi_M$ for the orthogonal projection onto $M$. The orthogonal projection onto $M^{\perp}$ is denoted by $(\Pi_M)_{\perp} \doteq I_H - \Pi_M = \Pi_{M^{\perp}}$. For functions $f, h \in L_2(X)$, $g \in L_2(Z)$, the rank-one operator $g \otimes f$ is defined as $(g \otimes f)(h) = \langle h, f \rangle_{L_2(X)} g$. This generalizes the standard outer product. For vectors $x \in \mathbb{R}^d$, we write $\|x\|_{\ell_2}$ for the Euclidean norm.

**Data Splitting and Empirical Expectations.** Given $n, m \geq 0$, we consider two independent datasets: an unlabeled dataset, $\tilde{\mathcal{D}}_m = \{(\tilde{z}_i, \tilde{x}_i)\}_{i=1}^m$, used to learn features for $X$ and $Z$, and a labeled dataset, $\mathcal{D}_n = \{(z_i, x_i, y_i)\}_{i=1}^n$, used to estimate the structural function. For $R \in \{X, Z, Y, (X, Z), (X, Y), (Z, Y), (X, Z, Y)\}$ and a measurable function $f$, we define the empirical expectation over $\tilde{\mathcal{D}}_m$ as $\hat{\mathbb{E}}_m[f(R)] \doteq \frac{1}{m} \sum_{i=1}^m f(\tilde{r}_i)$, and similarly for $\hat{\mathbb{E}}_n$ on $\mathcal{D}_n$.

**Feature Maps and Projection Operators.** Let $d \geq 1$, and let $\varphi : \mathcal{X} \to \mathbb{R}^d$ be a feature map with components $\varphi_i \in L_2(X)$ for $i = 1, \ldots, d$. We define $\mathcal{H}_{\varphi,d} \doteq \mathrm{span}\{\varphi_1, \ldots, \varphi_d\}$, and let $\Pi_{\varphi,d}$ be the $L_2-$orthogonal projection onto this space. Analogous definitions apply for feature maps $\psi : \mathcal{Z} \to \mathbb{R}^d$, yielding $\mathcal{H}_{\psi,d}$ and $\Pi_{\psi,d}$.

## 2.2 On the Conditional Expectation Operator: NPIV as an Ill-posed Inverse Problem

We begin by defining the conditional expectation operator. Let

$$\mathcal{T} : L_2(X) \to L_2(Z), \quad h \mapsto \mathbb{E}[h(X) \mid Z].$$

As highlighted by Eq. (2), this operator plays a central role in the NPIV model. It encodes the conditional distribution of $X$ given $Z$ and provides a convenient lens through which to analyze how efficiently one can estimate $h_0$. The NPIV equation admits a solution if and only if $r_0 \in \mathcal{R}(\mathcal{T})$, and is identifiable if and only if $\mathcal{T}$ is injective. We therefore adopt the following assumption.

**Assumption 1.** $\mathcal{T}$ *is injective and there exists a solution to the NPIV problem, i.e.,* $\mathcal{T}^{-1}(\{r_0\}) \neq \emptyset$.

The existence part is essential: without it, there is no structural function $h_0$ satisfying the model in Eq. (1). When $\mathcal{T}$ is not injective, we lose identifiability (*i.e.*, unicity of the solution). In such scenarios, it is possible to establish consistency to a minimum-norm solution of the inverse problem Eq. (2), namely $h_* \doteq \mathcal{T}^\dagger r_0$ [15].[3] We assume injectivity henceforth to simplify the discussion.

Eq. (2) is an inverse problem in which both the operator $\mathcal{T}$ and the right-hand side $r_0$ are unknown. This inverse problem is typically ***ill-posed***, as $\mathcal{T}^{-1}$ is generally not continuous.[4] Hence, even if $\mathcal{T}$ were known, an approximation $\hat{r} \approx r_0$ would not ensure that $\mathcal{T}^{-1}\hat{r}$ is close to $h_0 = \mathcal{T}^{-1}r_0$. In such settings, regularization is required to obtain stable approximate inverses. Two common regularization approaches are ***Tikhonov regularization***, where $\mathcal{T}^{-1} \approx (\mathcal{T}^\star \mathcal{T} + \lambda I)^{-1}\mathcal{T}^\star$, $\lambda > 0$, and ***spectral cut-off***, which truncates the Singular Value Decomposition (SVD) [14].

When $\mathcal{T}$ is compact (and therefore it admits a countable sequence of singular values), the ***degree of ill-posedness*** of the inverse problem is characterized by the decay rate of these singular values. Since the singular values of $\mathcal{T}^{-1}$ are the reciprocals of those of $\mathcal{T}$, faster decay implies worse ill-posedness. The NPIV problem can be viewed as particularly challenging, as in addition to the ill-posedness of the operator $\mathcal{T}$, one must also estimate it from data. We now introduce a sufficient condition for compactness of $\mathcal{T}$.

**Assumption 2.** *The joint distribution $\pi_{XZ}$ is dominated by the product measure $\pi_X \otimes \pi_Z$, and its Radon–Nikodym derivative, or "density", $p \doteq \frac{\mathrm{d}\pi_{XZ}}{\mathrm{d}\pi_X \otimes \pi_Z}$ belongs to $L_2(\mathcal{X} \times \mathcal{Z}, \pi_X \otimes \pi_Z)$.*

Assumption 2 is standard in the NPIV literature (Assumption A.1 in [11]). It may fail when $X$ and $Z$ share variables or are deterministically related. For example, if $Z = (W, V)$ and $X = V$, then the joint distribution is not dominated by the product measure.

**Proposition 1.** *Under Assumption 2, $\mathcal{T}$ is a Hilbert–Schmidt operator and thus compact.*

This result is well known in the inverse problems literature; we provide a proof in Appendix C.1.

**Compact Decomposition and Eigensubspaces.** As $\mathcal{T}$ is a conditional expectation operator, it is non-expansive, *i.e.*, $\|\mathcal{T}\|_{\mathrm{op}} \leq 1$. Under Assumption 2, $\mathcal{T}$ is compact, and therefore admits a SVD

$$\mathcal{T} = \sum_{i \geq 1} \sigma_i u_i \otimes v_i, \tag{3}$$

where $\{u_i\}$ and $\{v_i\}$ are orthonormal basis of $L_2(Z)$ and $L_2(X)$ respectively (left and right singular functions, respectively), and $\sigma_i$ are the nonnegative singular values in nonincreasing order. The leading singular triplet is known explicitly: $\sigma_1 = 1$, $u_1 = \mathbb{1}_{\mathcal{Z}}$ and $v_1 = \mathbb{1}_{\mathcal{X}}$ corresponding to the constant functions. Under the assumption that $\mathcal{T}$ is injective (Assumption 1) $\sigma_i > 0$ for all $i \geq 1$. Finally, since $\mathcal{T}$ is Hilbert–Schmidt, $\sum_{i \geq 1} \sigma_i^2 < \infty$. For any $d \geq 1$, define the leading eigensubspaces of $\mathcal{T}$ as

$$\mathcal{U}_d \doteq \mathrm{span}\{u_1, \ldots, u_d\}, \quad \Pi_{\mathcal{Z},d} : \text{ orthogonal projection onto } \mathcal{U}_d,$$
$$\mathcal{V}_d \doteq \mathrm{span}\{v_1, \ldots, v_d\}, \quad \Pi_{\mathcal{X},d} : \text{ orthogonal projection onto } \mathcal{V}_d.$$

---

[3]This corresponds to the unique element in the pre-image of $r_0$ under $\mathcal{T}$ that minimizes the $L_2$-norm and is equivalently characterized as the unique element of the set $\mathcal{T}^{-1}(\{r_0\}) \cap \mathcal{N}(\mathcal{T})^\perp$ [2].

[4]In particular, $\mathcal{T}^{-1}$ is discontinuous if $\mathcal{T}$ is compact and not finite rank.

In addition, define $\mathcal{T}_d \doteq \sum_{i=1}^d \sigma_i u_i \otimes v_i$ the rank-$d$ truncation of $\mathcal{T}$. To avoid ambiguity in the definition of $\mathcal{U}_d$, $\mathcal{V}_d$ and $\mathcal{T}_d$, we assume that $\sigma_d > \sigma_{d+1}$. We refer to Appendix A for more details on spectral theory.

## 3 Solving NPIV with 2SLS

To understand how and when spectral features should be used, we first recall results from the Sieve 2SLS literature. This gives us a clear picture of what terms are to be controlled in order to achieve good generalization, which we will use to demonstrate the effectiveness of spectral features and reveal our good-bad-ugly taxonomy.

### 3.1 Sieve 2SLS estimator

We start with the following characterization of the structural function:

$$h_0 = \underset{h \in L_2(X)}{\arg\min} \, \mathbb{E}\left[(Y - \mathcal{T}h(Z))^2\right],$$

where the minimizer, $h_0$, is unique under Assumption 1. A popular strategy to estimate $h_0$ is to consider a hypothesis space $\mathcal{H}_{\mathcal{X}} \subset L_2(X)$ and a procedure to estimate the action of $\mathcal{T}$: $\hat{\mathcal{T}}h \approx \mathcal{T}h$ for all $h \in \mathcal{H}_{\mathcal{X}}$, in order to obtain

$$\hat{h} \in \underset{h \in \mathcal{H}_{\mathcal{X}}}{\arg\min} \, \hat{\mathbb{E}}_n\left[\left(Y - \hat{\mathcal{T}}h(Z)\right)^2\right], \tag{4}$$

possibly subject to regularization. In the algorithm 2SLS, we consider features on both $\mathcal{X}$ and $\mathcal{Z}$, and both steps (estimating $h \mapsto \mathcal{T}h$ and solving Eq. (4)) are carried out via linear regression in feature space. The two sets of features may be infinite-dimensional—*e.g.*, using Reproducing Kernel Hilbert Spaces (RKHS)—or finite-dimensional—*e.g.*, using splines, wavelets, or neural network features. When the feature dimension grows with sample size, we obtain a **sieve**. Since our focus is on neural network-based features, we present the finite-dimensional version, noting that the ideas naturally extend to infinite-dimensional cases with proper regularization [35, 31]. We refer to Appendix D for a description of other 2SLS methods.

Consider a feature map $\varphi : \mathcal{X} \to \mathbb{R}^d$ such that $\mathcal{H}_{\mathcal{X}} \doteq \{x \mapsto \theta^\mathsf{T}\varphi(x), \theta \in \mathbb{R}^d\}$ is included in $L_2(X)$. Then for any $h = \theta^\mathsf{T}\varphi(\cdot) \in \mathcal{H}_{\mathcal{X}}$, $\mathcal{T}h(Z) = \theta^\mathsf{T}\mathbb{E}[\varphi(X) \mid Z]$. Therefore, estimating $F_\star(Z) \doteq \mathbb{E}[\varphi(X) \mid Z]$ allows us to estimate $\mathcal{T}h$ for any $h \in \mathcal{H}_{\mathcal{X}}$. Let $\psi : \mathcal{Z} \to \mathbb{R}^d$ be another feature map with $\mathcal{H}_{\mathcal{Z}} \doteq \{z \mapsto \alpha^\mathsf{T}\psi(z), \alpha \in \mathbb{R}^d\} \subset L_2(Z)$. We estimate $F_\star$ with vector-valued linear regression in feature space by approximating $F_\star$ with $A\psi(\cdot)$, where $A \in \mathbb{R}^{d \times d}$, solving

$$\hat{A} \doteq \hat{\mathbb{E}}_n[\varphi(X)\psi(Z)^\mathsf{T}]\hat{\mathbb{E}}_n[\psi(Z)\psi(Z)^\mathsf{T}]^\dagger \in \underset{A \in \mathbb{R}^{d \times d}}{\arg\min} \, \hat{\mathbb{E}}_n[\|\varphi(X) - A\psi(Z)\|_{\ell_2}^2].$$

The pseudo-inverse yields the **minimum-norm solution** in the case of non-uniqueness. Regularized versions, such as Tikhonov-regularized least squares (also known as ridge regression), can be obtained by penalizing with $\eta\|A\|_{\mathrm{HS}}^2$ for $\eta > 0$. Computing $\hat{A}$ is referred to as the **first stage** of the 2SLS algorithm. For the **second stage**, for any $h = \theta^\mathsf{T}\varphi(\cdot) \in \mathcal{H}_{\mathcal{X}}$, we define $\hat{\mathcal{T}}h(Z) \doteq \theta^\mathsf{T}\hat{F}(Z)$ with $\hat{F}(Z) = \hat{A}\psi(Z)$. Plugging this into Eq. (4) yields $\hat{h} = \hat{\theta}^\mathsf{T}\varphi(\cdot)$ with

$$\hat{\theta} \doteq \hat{\mathbb{E}}_n[\hat{F}(Z)\hat{F}(Z)^\mathsf{T}]^\dagger \hat{\mathbb{E}}_n[\hat{F}(Z)Y] \in \underset{\theta \in \mathbb{R}^d}{\arg\min} \, \hat{\mathbb{E}}_n[(Y - \theta^\mathsf{T}\hat{F}(Z))^2]. \tag{5}$$

Similarly to the first stage, we can also consider a regularized version of the second stage, where we penalize with $\lambda\|\theta\|_{\ell_2}^2$ for $\lambda > 0$. $\hat{h}$ is the sieve NPIV estimator [3, 7, 6].

**2SLS versus saddle-point estimator.** Based on features $\varphi$ and $\psi$, [43, 36] proposed an estimator in a saddle-point form: $\hat{h}_{bis} = \hat{\theta}_{bis}^\mathsf{T}\varphi(\cdot)$, with

$$\hat{\theta}_{bis} = \underset{\theta \in \mathbb{R}^d}{\arg\min} \, \underset{\alpha \in \mathbb{R}^d}{\max} \, \hat{\mathbb{E}}_n[\alpha^\mathsf{T}\psi(Z)(Y - \theta^\mathsf{T}\varphi(X)) - 0.5 \cdot (\alpha^\mathsf{T}\psi(Z))^2] + \lambda\|\theta\|_{\ell_2}^2.$$

Albeit looking different, solving this min-max problem in closed-form leads to $\hat{\theta}_{bis} = \hat{\theta}$ (when we introduce regularization in Eq. (5)), and thus $\hat{h}_{bis} = \hat{h}$. The saddle-point form could be useful for deriving convergence rates, see [2] for example, as it allows us to use the theory of saddle-point problems. However, it is not necessary for the implementation of the algorithm.

## 3.2 Convergence rates for Sieve 2SLS

In order to present convergence rates for Sieve 2SLS, we introduce the sieve measure of ill-posedness.

**Definition 1.** *Given $d \geq 1$ and $\mathcal{H}_{\varphi,d} = \mathrm{span}\{\varphi_1, \ldots, \varphi_d\}$, the sieve measure of ill-posedness is defined as*

$$\tau_{\varphi,d} \doteq \sup_{h \in \mathcal{H}_{\varphi,d},\, h \neq 0} \frac{\|h\|_{L_2(X)}}{\|\mathcal{T}h\|_{L_2(Z)}} = \left( \inf_{h \in \mathcal{H}_{\varphi,d},\, \|h\|_{L_2}=1} \|\mathcal{T}h\|_{L^2(Z)} \right)^{-1}.$$

This quantity can be interpreted as the operator norm of $\mathcal{T}^{-1}$ restricted to $\mathcal{T}(\mathcal{H}_{\varphi,d})$; that is, $\tau_{\varphi,d} = \sup_{g \in \mathcal{T}(\mathcal{H}_{\varphi,d}),\, h \neq 0} \|\mathcal{T}^{-1}g\|_{L_2(X)}/\|g\|_{L_2(Z)}$. While the full operator $\mathcal{T}^{-1}$ is typically unbounded, the restricted norm $\tau_{\varphi,d}$ remains finite. It captures the degree of ill-posedness specific to the chosen hypothesis space $\mathcal{H}_{\varphi,d}$. As $d$ increases and the hypothesis class becomes richer, $\tau_{\varphi,d}$ increases. Equipped with this quantity, we now introduce stability conditions that will allow us to establish generalization bounds for Sieve 2SLS.

**Assumption 3.** *(i) $\sup_{h \in \mathcal{H}_{\varphi,d},\, \|h\|_{L_2}=1} \left\| (\Pi_{\psi,d})_\perp \mathcal{T}h \right\|_{L^2(Z)} = o_d\left( \tau_{\varphi,d}^{-1} \right)$; (ii) there is a constant $C > 0$ such that $\left\| \mathcal{T} (\Pi_{\varphi,d})_\perp h_0 \right\|_{L^2(Z)} \leq C \times \tau_{\varphi,d}^{-1} \times \left\| (\Pi_{\varphi,d})_\perp h_0 \right\|_{L^2(X)}$.*

Assumption 3-(i) is a condition on the approximation properties of the features used for the instrument space. It says the image of $\mathcal{H}_{\varphi,d}$ through $\mathcal{T}$ lies almost entirely inside $\mathcal{H}_{\psi,d}$, *i.e.*, the projection onto the orthogonal complement vanishes faster than the inverse ill-posedness. Assumption 3-(ii) is a stability condition sometimes referred as **link condition** that is standard in the NPIV literature (see Assumption 6 in [3], Assumption 5.2(ii) in [7] and Assumption 3 in [23]). It says that $\mathcal{T}$ becomes increasingly contractive on the component of $h_0$ lying outside the sieve space as the sieve dimension increases. The next assumption ensures the conditional variance of the noise is uniformly bounded.

**Assumption 4.** *There exists $\bar{\sigma} > 0$ such that $\mathbb{E}\left[ U^2 \mid Z \right] \leq \bar{\sigma}^2 < \infty$ almost everywhere.*

The following result is an upper bound on the generalization error from [6].

**Theorem 1** (Theorem B.1. [6]). *Let $\hat{h}$ be the 2SLS estimator from Eq. (5), using features $\varphi : \mathcal{X} \to \mathbb{R}^d$ and $\psi : \mathcal{Z} \to \mathbb{R}^d$. Suppose Assumptions 1, 3 and 4 hold and $\tau_{\varphi,d}\zeta_{\varphi,\psi,d}\sqrt{(\log d)/n} = o(1)$, where $\zeta_{\varphi,\psi,d} \doteq \max\{\sup_x \left\| \mathbb{E}[\varphi(X)\varphi(X)^\intercal]^{-1/2}\varphi(x) \right\|_{\ell_2},\, \sup_z \left\| \mathbb{E}[\psi(Z)\psi(Z)^\intercal]^{-1/2}\psi(z) \right\|_{\ell_2}\}$. Then:*

$$\left\| \hat{h} - h_0 \right\|_{L^2(X)} = O_p\left( \|(\Pi_{\varphi,d})_\perp h_0\|_{L^2(X)} + \tau_{\varphi,d}\sqrt{d/n} \right).$$

The first term captures the **approximation error**, while the second term corresponds to the **estimation error** arising from finite samples. When standard bases such as cosine, spline, or wavelet functions are used, the approximation error is well-understood. If the sieve ill-posedness $\tau_{\varphi,d}$ grows at a polynomial or exponential rate in $d$, one can choose $d = d(n)$ to balance the two terms optimally [3]. Under such growth conditions and smoothness assumptions on $h_0$ (*e.g.*, $h_0$ belongs to a Sobolev ball), the resulting convergence rate is minimax optimal [8].

## 3.3 Convergence rates for Sieve 2SLS with spectral features

A drawback of using non-adaptive features (such as cosine or spline bases) is that verifying Assumption 3 or characterizing the growth of $\tau_{\varphi,d}$ is generally difficult. In contrast, we now consider features constructed from the SVD of $\mathcal{T}$, for which both conditions can be exactly characterized. Moreover, while estimators based on universal bases can achieve minimax optimality under smoothness assumptions on $\mathcal{T}$ and $h_0$, they are often outperformed in practice by estimators using data-dependent features. This highlights that minimax rates, though important, do not capture the full picture of empirical performance. Recall from Eq. (3) the singular value decomposition of $\mathcal{T}$, and the definition of the top-$d$ right eigenspace $\mathcal{V}_d$. The following result shows that this subspace achieves the minimal possible measure of ill-posedness among all $d$-dimensional subspaces of $L_2(X)$.

**Proposition 2** (Lemma 1 [3]). *Let Assumption 2 hold. For any $d \geq 1$, the smallest possible value of $\tau_{\varphi,d}$ is $\tau_{\varphi,d} = \sigma_d^{-1}$, achieved when $\mathcal{H}_{\varphi,d} = \mathcal{V}_d$.*

**Definition 2** (Spectral features). *We say that the features $\varphi$ and $\psi$ are **spectral features** if*

$$\mathcal{H}_{\varphi,d} = \mathrm{span}\{v_1, \ldots, v_d\} = \mathcal{V}_d \quad and \quad \mathcal{H}_{\psi,d} = \mathrm{span}\{u_1, \ldots, u_d\} = \mathcal{U}_d.$$

By Proposition 2, spectral features minimize the sieve measure of ill-posedness, and therefore the estimation error term. This leads to the following corollary.

**Corollary 1** (Sieve 2SLS with spectral features). *Let $\hat{h}$ be the 2SLS estimator from Eq.* (5) *using spectral features. Let Assumptions 1, 2 and 4 hold and $\sigma_d^{-1}\zeta_{\varphi,\psi,d}\sqrt{(\log d)/n} = o(1)$. Then:*

$$\left\|\hat{h} - h_0\right\|_{L^2(X)} = O_p\left(\|(\Pi_{\mathcal{X},d})_\perp h_0\|_{L^2(X)} + \sqrt{\frac{d}{n\sigma_d^2}}\right).$$

We postpone the proof to Appendix C.2.

**The good, the bad, and the ugly.** The bound in Corollary 1 decomposes the generalization error into two terms: ***spectral alignment***, encoded in the approximation error $\|(\Pi_{\mathcal{X},d})_\perp h_0\|$, which measures how well $h_0$ lies in the top-$d$ singular space; and ***ill-posedness***, captured by the inverse singular value $\sigma_d^{-1}$. In the ***good*** regime, $h_0$ lies mostly in $\mathcal{U}_d$ and $\sigma_d$ decays slowly, so both terms are small. In the ***bad*** regime, alignment is favorable but fast spectral decay inflates the estimation error. In the ***ugly*** regime, $h_0$ is misaligned with the top eigenspaces, so the approximation error dominates and estimation fails regardless of $\sigma_d$ or $n$. In Appendix B, we show that the bound in Corollary 1 is tight under a source condition and a singular value decay condition.

## 4 Sieve 2SLS with learned spectral features

In the previous section, we showed that spectral features minimize the sieve measure of ill-posedness and lead to fast generalization rates when the structural function $h_0$ aligns well with the top eigenspaces of the conditional expectation operator $\mathcal{T}$. This motivates learning such features directly from data, especially in the case where the conditional distribution of $X$ given $Z$ is informative (*i.e.*, the instrument is strong) and the alignment with the spectrum of $\mathcal{T}$ is favorable. We now demonstrate that the contrastive learning approach of [36] can be equivalently seen as explicitly targeting the leading eigenspaces of $\mathcal{T}$. Consider the following objective:

$$\mathcal{L}_d(\varphi,\psi) \doteq \|\mathcal{T}_d(\varphi,\psi) - \mathcal{T}\|_{\mathrm{HS}}^2, \qquad \mathcal{T}_d(\varphi,\psi) \doteq \sum_{i=1}^d \psi_i \otimes \varphi_i, \tag{6}$$

where $\varphi_i \in L_2(X)$, $\psi_i \in L_2(Z)$, $i = 1,\ldots,d$. As any rank-$d$ operator can be decomposed as $\mathcal{T}_d(\varphi,\psi)$, we have the following version of the Eckart–Young–Mirsky Theorem for the Hilbert–Schmidt norm:

**Theorem 2** (Eckart–Young–Mirsky Theorem). *Let Assumption 2 hold and let $d \geq 1$ be such that $\sigma_d > \sigma_{d+1}$. For all feature maps $\varphi : \mathcal{X} \to \mathbb{R}^d$, $\psi : \mathcal{Z} \to \mathbb{R}^d$, we have*

$$\mathcal{L}_d(\varphi,\psi) \geq \|\mathcal{T} - \mathcal{T}_d\|_{\mathrm{HS}}^2 = \sum_{i>d} \sigma_i^2,$$

*where equality holds if and only if $\mathcal{T}_d(\varphi,\psi) = \mathcal{T}_d$, in which case $\mathcal{H}_{\varphi,d} = \mathcal{V}_d$, $\mathcal{H}_{\psi,d} = \mathcal{U}_d$. $\mathcal{T}_d$ is the rank-$d$ truncation of $\mathcal{T}$ introduced in Section 2.*

While the Eckart–Young–Mirsky objective in Eq. (6) offers a clean objective to learn spectral features, it appears impractical due to the unknown nature of the operator $\mathcal{T}$. However, this objective admits an equivalent formulation as a spectral contrastive learning loss [37]:

$$\mathcal{L}_d(\varphi,\psi) = \mathbb{E}_X\mathbb{E}_Z\left[(\varphi(X)^\mathsf{T}\psi(Z))^2\right] - 2\mathbb{E}_{X,Z}\left[\varphi(X)^\mathsf{T}\psi(Z)\right] + \|\mathcal{T}\|_{\mathrm{HS}}^2, \tag{7}$$

where the last term is independent of $(\varphi,\psi)$. We provide a proof of this equivalence in Appendix C.3. The population loss can be estimated from samples $\tilde{\mathcal{D}}_m = \{(\tilde{x}_i, \tilde{z}_i)\}_{i=1}^m$ as

$$\hat{\mathcal{L}}_d(\varphi,\psi) \doteq \frac{1}{m(m-1)} \sum_{i \neq j} (\varphi(\tilde{x}_i)^\mathsf{T}\psi(\tilde{z}_j))^2 - \frac{2}{m} \sum_{i=1}^m \varphi(\tilde{x}_i)^\mathsf{T}\psi(\tilde{z}_i). \tag{8}$$

The spectral contrastive loss is the basis of the recent algorithm proposed in [36], where $\varphi$ and $\psi$ are parametrized by neural networks and optimized via stochastic gradient descent. While [36] motivate this objective heuristically, Eq. (6) shows that it directly targets the leading singular structure of $\mathcal{T}$. The use of objectives similar to Eq. (6) for operator learning or Eq. (7) for representation learning has a rich history that we detail in Appendix D.

**Comparison to [36].** To justify the use of the contrastive loss in Eq. (8), [36] assume that the density $p(x, z)$ factorizes as

$$p(x, z) = \varphi(x)^\intercal \psi(z), \quad \varphi : \mathcal{X} \to \mathbb{R}^d, \quad \psi : \mathcal{Z} \to \mathbb{R}^d. \tag{9}$$

We argue that this assumption is overly strong: it holds if and only if $\mathcal{T}$ has rank at most $d$, in which case the spectral features perfectly capture the structure of $\mathcal{T}$. This result is formalized in Proposition 5, Appendix C.3. We should instead interpret Eq. (9) as an approximation rather than a literal assumption. In practice, the learned features $\varphi, \psi$ provide a rank-$d$ approximation of $\mathcal{T}$, and the quality of this approximation governs the performance of the resulting Sieve 2SLS estimator.

Let $\hat{\varphi}, \hat{\psi}$ be $d$-dimensional features trained with loss Eq. (8) with $\tilde{D}_m$ and define $\hat{\mathcal{T}}_d \doteq \sum_{i=1}^d \hat{\psi}_i \otimes \hat{\varphi}_i$. We make the following assumption:

**Assumption 5.** $\{\hat{\psi}_1, \ldots, \hat{\psi}_d\}$ and $\{\hat{\varphi}_1, \ldots, \hat{\varphi}_d\}$ form linearly independent families.

This assumption is only made to simplify the discussion. If $\{\hat{\psi}_1, \ldots, \hat{\psi}_d\}$ or $\{\hat{\varphi}_1, \ldots, \hat{\varphi}_d\}$ is not linearly independent, one can replace $d$ in the following by the linear dimension of the family. Consider $\sigma_d > \sigma_{d+1}$ so that the unique minimizer of the population loss is given by $\mathcal{T}_d$. We quantify how the distance between $\hat{\mathcal{T}}_d$ and $\mathcal{T}_d$ affects the approximation error and the sieve measure of ill-posedness $\tau_{\hat{\varphi},d}$. We quantify how the distance between $\hat{\mathcal{T}}_d$ and $\mathcal{T}_d$ affects the generalization error of the 2SLS estimator.

**Theorem 3.** *Let Assumptions 1, 2, 4 and 5 hold and let* $\hat{\varepsilon}_d \doteq \left\| \hat{\mathcal{T}}_d - \mathcal{T}_d \right\|_{\mathrm{op}}$ *be such that* $\hat{\varepsilon}_d < (1 - 1/\sqrt{2})\sigma_d$.

  *i)* $\sigma_d^{-1} \leq \tau_{\hat{\varphi},d} \leq (\sigma_d - 2\hat{\varepsilon}_d)^{-1}$;
  *ii) Let* $\hat{h}$ *be the 2SLS estimator from Eq. (5), using features* $\hat{\varphi} : \mathcal{X} \to \mathbb{R}^d$ *and* $\hat{\psi} : \mathcal{Z} \to \mathbb{R}^d$. *Suppose* $(\sigma_d - 2\hat{\varepsilon}_d)^{-1} \zeta_{\hat{\varphi},\hat{\psi},d} \sqrt{(\log d)/n} = o(1)$ *and* $\hat{\varepsilon}_d = o_d(\sigma_d^2)$, *then*

$$\left\| \hat{h} - h_0 \right\|_{L^2(X)} = O_p \left( \|(\Pi_{\mathcal{X},d})_\perp h_0\|_{L_2(X)} + \frac{\sqrt{2}\hat{\varepsilon}_d \|h_0\|_{L_2(X)}}{\sigma_d} + \frac{1}{\sigma_d - 2\hat{\varepsilon}_d} \sqrt{\frac{d}{n}} \right).$$

We postpone the proof to Appendix C.3. The result quantifies how close the learned features must be to $\mathcal{T}_d$ in order to preserve the desirable properties of spectral 2SLS. Theorem 3-i) shows that as long as the learned operator $\hat{\mathcal{T}}_d$ is a good approximation of the true rank-$d$ truncation $\mathcal{T}_d$, the learned features inherit a favorable measure of ill-posedness. Theorem 3-ii) further ensures that if contrastive learning yields a near-optimal rank-$d$ approximation of $\mathcal{T}$, the downstream Sieve 2SLS estimator retains strong statistical guarantees. Importantly, the spectral features generalization error $\hat{\varepsilon}_d$ encapsulates the statistical and optimization errors in learning spectral features. It depends on the number of samples used to train the contrastive loss, the expressivity and architecture of the neural networks parametrizing $\varphi$ and $\psi$, and the complexity of the leading eigensubspaces of $\mathcal{T}$. As $\mathcal{L}_d(\varphi, \psi)$ converges to $\mathcal{L}_d^{\min} \doteq \|\mathcal{T} - \mathcal{T}_d\|_{\mathrm{HS}}^2$, $\hat{\varepsilon}_d$ converges to 0 by Theorem 2. Precisely controlling $\hat{\varepsilon}_d$ lies beyond the scope of this paper and remains an important question for future work.

## 5 Estimating alignment and spectral decay from data

Having established the good-bad-ugly taxonomy based on spectral properties and its interaction with the structural function $h_0$, we now discuss how to estimate these spectral properties from data. Namely, we present a methodology to estimate the spectral decay of the operator $\mathcal{T}$ as well as spectral alignment with the structural function $h_0$. Recall that $\mathcal{T}$ is compact and admits an SVD of the form Eq. (3), with the key relationship that $\sigma_i v_i = \mathcal{T}^* u_i$. Therefore, for all $i \geq 1$, we have

$$\langle v_i, h_0 \rangle_{L_2(X)} = \frac{1}{\sigma_i} \langle \mathcal{T}^* u_i, h_0 \rangle_{L_2(X)} = \frac{1}{\sigma_i} \langle u_i, r_0 \rangle_{L_2(Z)} = \frac{1}{\sigma_i} \mathbb{E}[Y \cdot u_i(Z)], \tag{10}$$

where we used $\mathcal{T}h_0 = r_0$, the definition of $r_0$ and the tower property of the conditional expectation. This relationship shows that the alignment coefficients $\langle v_i, h_0 \rangle_{L_2(X)}$ can be estimated from data if we have estimates of the singular functions $u_i$ and singular values $\sigma_i$. In Section 5, we provide a practical estimator for Eq. (10) using learned features and derive its estimation guarantees, which depend on the operator error $\hat{\varepsilon}_d$.

We apply this procedure in Section 6 to diagnose the spectral properties of the dSprites dataset [30], a popular benchmark in the nonparametric IV and proxy literature [40, 41, 36].

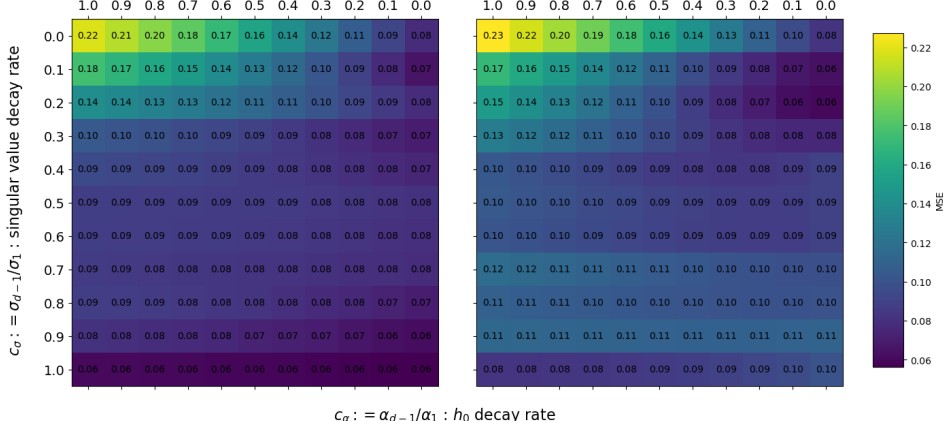

Figure 1: Dependence of the MSE of instrumental variable regression on the decay rates of the spectrum and coefficients of the structural function. IV fitting and the MSE evaluations were repeated 500 times per parameter set, rendering the standard error of these estimates negligible. **Left:** Oracle spectral features; **Right:** Learned features.

## 6 Experiments

### 6.1 Synthetic datasets

We evaluate the main theoretical insight of the paper: Sieve 2SLS with spectral features performs best when $h_0$ aligns with the top eigenspaces of $\mathcal{T}$ and its singular values decay slowly. Performance degrades with faster decay and weaker alignment. We design a synthetic NPIV setting where we control the conditional expectation operator $\mathcal{T}$ and its spectral decay and well as the alignment of $h_0$ with its eigenspaces. To simulate such a setting, we rely on the following procedure for generating samples from the NPIV model:

**Proposition 3.** *Let $\pi_{XZ}$ be a probability distribution on $\mathcal{X} \times \mathcal{Z}$ with marginals $\pi_X$ and $\pi_Z$. Let $\mathcal{T} : L_2(\mathcal{X}, \pi_X) \to L_2(\mathcal{Z}, \pi_Z)$ be the conditional expectation operator associated with $\pi_{XZ}$ and let $h_0 \in L_2(\mathcal{X}, \pi_X)$. Sample $(X, Z) \sim \pi_{XZ}$, $V \sim \mathcal{N}(0, \sigma^2)$ and set $U = \mathcal{T}h_0(Z) - h_0(X) + V$ and $Y = h_0(X) + U$. Then $(U, Z, X, Y)$ is a sample from the NPIV model of Eq. (1).*

Let $d = 11$, $\mathcal{X} = [0, 2\pi]$, $\mathcal{Z} = [0, 2\pi]$, $\sigma^2 = 0.1$, $f(x) = (\sin(x), \sin(2x), \ldots, \sin((d-1)x))$, $g(z) = (\sin(z), \sin(2z), \ldots, \sin((d-1)z))$, and let $P, Q$ be two orthogonal $(d-1) \times (d-1)$ matrices. Define $v : \mathcal{X} \to \mathbb{R}^{d-1}$ and $u : \mathcal{Z} \to \mathbb{R}^{d-1}$ as $v(x) = Pf(x)$ and $u(z) = Qg(z)$. We set $\mathcal{T} = \mathbb{1}_{\mathcal{Z}} \otimes \mathbb{1}_{\mathcal{X}} + \sum_{i=1}^{d-1} \sigma_i u_i \otimes v_i$. $(X, Z)$ is jointly sampled with rejection sampling. $Z, X$ are uniform on $[0, 2\pi]$ and $X$ given $Z$ admits $\mathcal{T}$ as conditional expectation operator. We then sample $Y$ following Proposition 3. The multiplication of the trigonometric functions by the orthogonal matrices $P, Q$ ensures that the consecutive singular functions of $\mathcal{T}$ are comparably complex. By keeping the difficulty constant per singular function we can isolate the influence of the singular value decay on the performance of the final estimator. We set the singular values $\{\sigma_i\}_{i=1}^{d-1}$ to decay linearly, from some initial value $\sigma_1$ to a $\sigma_{d-1} = c_\sigma \sigma_1$, $c_\sigma \in [0, 1]$. To control spectral alignment, we set $h_0 = \sum_{i=1}^{d-1} \alpha_i v_i$ subject to $\|\alpha\|_{\ell_2} = 1$, and allow $\{\alpha_i\}_{i=1}^{d-1}$ to decay linearly from $\alpha_1$ to $\alpha_{d-1} = c_\alpha \alpha_1$, $c_\alpha \in [0, 1]$. Each value of $c_\alpha \in [0, 1]$ controls the alignment of $h_0$ with the top singular functions of $\mathcal{T}$: larger values correspond to weaker alignment. Similarly, $c_\sigma \in [0, 1]$ controls the ill-posedness of the problem: smaller values indicate faster decay of singular values and hence more severe ill-posedness.

In Figure 1, we report the mean squared error (MSE) of two 2SLS estimators across a range of spectral decay rates ($c_\sigma$) and structural function decay rates ($c_\alpha$). The left panel uses 2SLS with fixed features $f$ and $g$ that form the exact spectral features. This represents an oracle setting. The MSE decreases from top to bottom as the problem becomes better conditioned (i.e., as $c_\sigma$ increases). Additionally, the MSE decreases from left to right, as smaller values of $c_\alpha$ correspond to stronger

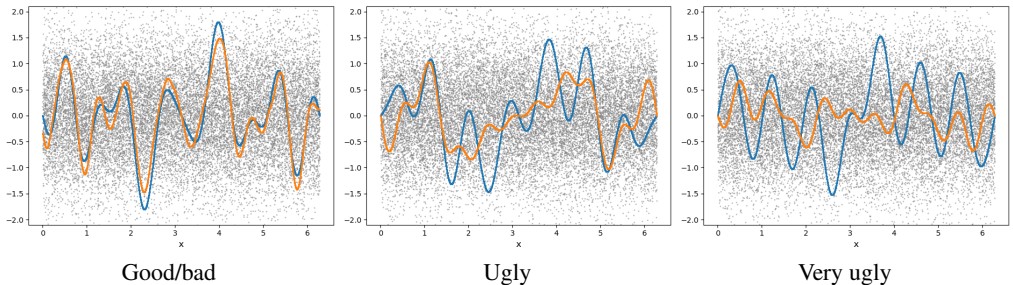

| Good/bad | Ugly | Very ugly |

Figure 2: Comparison of the spectral methods' performance depending on the alignment of $h_0$ with the singular functions of $\mathcal{T}$. **Blue:** True $h_0$; **Orange:** 2SLS estimate.

spectral alignment: more of $h_0$ is concentrated on the top singular functions of $\mathcal{T}$. These trends are fully consistent with the bound established in Corollary 1. In the right panel, we learn 50 features from data by minimizing the empirical contrastive loss Eq. (8) using a neural network. We then build a 2SLS estimator on top of the learned features. The same qualitative trends are observed: MSE decreases with slower singular value decay and better alignment. Moreover, when $c_\sigma$ is small (top rows), learning the entire eigenspaces becomes difficult. In this regime, performance is most sensitive to $c_\alpha$. In particular, if $h_0$ is concentrated on the top singular functions (small $c_\alpha$), good recovery is still possible. In contrast, when $c_\sigma$ is large (bottom rows), the singular functions are equally important and more easily learned, rendering the alignment of $h_0$ less critical. The mismatch between the oracle and learned settings is because we do not exactly recover the singular feature spaces; as predicted by Theorem 3. It is also worth noting that when $h_0$ is concentrated on high singular values, it may sometimes be beneficial for the singular values to decay quickly, by making it easier for the feature-learning model to learn the top singular functions well. This may explain the slight top-bottom increase in losses on the right end of the right panel.

The previous experiment assumes $h_0$ lies in the span of the singular functions of $\mathcal{T}$. Figure 2 illustrates three regimes. The left panel shows a well-aligned case ("good/bad"), where $h_0$ is entirely supported on the singular functions of $\mathcal{T}$, allowing for exact recovery as long as enough data is available. The middle panel represents a "partial alignment" regime, where $h_0$ has some overlap with the singular functions; in this case, recovery is imperfect, but the estimator can still capture some signal. Finally, the right panel illustrates the "misaligned" or "very ugly" regime, where $h_0$ is orthogonal to the singular functions. The instrument provides no information about the signal, and all spectral methods fail. We refer to Appendix E for further discussion of these scenarios and the experimental setup.

### 6.2 Dsprites dataset

The dSprites dataset, as employed in the evaluation of IV models, consists of $64 \times 64$ noisy images of hearts, with varying $x, y$ positions, orientations and sizes. The full setting is as follows; $X \in \mathbb{R}^{64 \times 64}$ is the raw image, $Z \in \mathbb{R}^3$ consists of the $x$ position, orientation and scale of the heart in the image, and the output is constructed as[5]

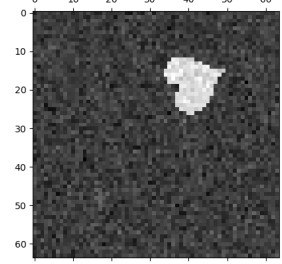

$$Y = h_0(X) + 32 \cdot (y \text{ position} - 32) + U, \quad U \sim \mathcal{N}(0, 1),$$

where

$$h_0(X) = (\|A \circ X\|^2 - 3000)/500, \quad A_{ij} = |32 - j|/32.$$

Figure 3: Example image from the dSprites dataset.

It was observed by Sun et al. [36] that spectral methods significantly outperform alternatives in this setting. We argue that it is because the structural function employed in this benchmark lies in the "good" regime. Utilising the alignment estimation methods outlined in Section 5, we are able to provide evidence for this claim. We observe that the structural function is spanned by the leading singular functions of the conditional expectation operator associated with this model. The details of this experiment can be found in Appendix F.

---

[5]The original formulation of the dSprites model for IV used $A_{i,j} \sim \mathcal{U}(0, 1)$ [40]. However, it is noted in [41] (arxiv version 18/06/2024, Appendix E.3) that it leads to identifiability issues and that the choice $A_{ij} = |32 - j|/32$ is preferred.

## Acknowledgements

D.M., J.W., and A.G. are supported by the Gatsby Charitable Foundation. V.R.K. is supported from NextGenerationEU and MUR PNRR project PE0000013 CUP J53C22003010006 "Future Artificial Intelligence Research (FAIR)". A.M. has received funding from the European Research Council (ERC), under the European Union's Horizon 2020 research and innovation programme (Grant agreement No. 950180), and travel support from ELSA (Project ID: 101070617).

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

# Supplementary Material: Demystifying Spectral Feature Learning for Instrumental Variable Regression

## A    Background on Spectral Theory

For completeness, we briefly recall key notions from spectral theory used throughout the paper. Our goal is to provide intuition and the minimal mathematical tools required to interpret concepts such as compactness, singular value decomposition (SVD) of operators, and eigensubspaces.

Let $H_1$ and $H_2$ be separable Hilbert spaces with inner products $\langle \cdot, \cdot \rangle_{H_i}$ and norms $\| \cdot \|_{H_i}$, $i = 1, 2$.

**Hilbert spaces and bounded operators.**    A linear map $A : H_1 \to H_2$ is said to be ***bounded*** if there exists $C > 0$ such that $\|Ah\|_{H_2} \leq C\|h\|_{H_1}$ for all $h \in H_1$. The smallest such constant is the ***operator norm***, denoted $\|A\|_{\mathrm{op}}$.

**Adjoint and self-adjoint operators.**    Every bounded operator $A : H_1 \to H_2$ admits an ***adjoint*** $A^* : H_2 \to H_1$ defined by $\langle Ah, g \rangle_{H_2} = \langle h, A^*g \rangle_{H_1}$. If $H_1 = H_2$ and $A = A^*$, the operator is ***self-adjoint***.

**Compact operators.**    A linear operator $A$ is called ***compact*** if it maps bounded sets in $H_1$ to relatively compact sets in $H_2$, that is, the image of any bounded sequence has a convergent subsequence in $H_2$. All compact operators are bounded, but not all bounded operators are compact. Any compact operator is a limit (in operator norm) of finite-rank operators, so that the class of compact operators can be defined alternatively as the closure of the set of finite-rank operators in the operator norm topology. The conditional expectation operator $\mathcal{T}$ is a canonical example of a compact operator (see Proposition 1 below).

**Hilbert–Schmidt operators and norm.**    Let $A : H_1 \to H_2$ be a bounded linear operator. Choose any orthonormal basis $\{e_j\}_{j \geq 1}$ of $H_1$. The operator $A$ is called ***Hilbert–Schmidt*** if

$$\|A\|_{\mathrm{HS}}^2 := \sum_{j \geq 1} \|Ae_j\|_{H_2}^2 < \infty.$$

The value of the sum above is independent of the chosen basis of $H_1$. Hence the quantity $\langle A, B \rangle_{\mathrm{HS}} := \sum_{j \geq 1} \langle Ae_j, Be_j \rangle_{H_2}$ defines an inner product on the space of Hilbert–Schmidt operators, and $\| \cdot \|_{\mathrm{HS}}$ is the induced norm.

**Singular Value Decomposition (SVD).**    When $A : H_1 \to H_2$ is compact (not necessarily self-adjoint), it admits a ***singular value decomposition***:

$$A = \sum_{i \geq 1} \sigma_i\, u_i \otimes v_i,$$

where $\sigma_1 \geq \sigma_2 \geq \cdots \geq 0$ are the singular values of $A$ such that $\sigma_i \to 0$, and $\{v_i\} \subset H_1$, $\{u_i\} \subset H_2$ form orthonormal bases satisfying $Av_i = \sigma_i u_i$ and $A^* u_i = \sigma_i v_i$. This generalizes the matrix SVD to infinite-dimensional spaces. Important properties we use below:

- The sum above is finite if and only if $A$ has finite rank.
- $\|A\|_{\mathrm{op}} = \sigma_1$.
- If $A$ is Hilbert–Schmidt then $A$ is compact such that $\|A\|_{\mathrm{HS}}^2 = \sum_i \sigma_i^2 < \infty$, hence $\|A\|_{\mathrm{op}} \leq \|A\|_{\mathrm{HS}}$. Thus Hilbert–Schmidt operators are bounded, but the converse need not hold.

## B    Learning Rate with Spectral Features

In this section, we formalize the learning rate that can be obtained from Corollary 1. To do so, we employ two standard assumptions from the nonparametric IV literature [8]: a source condition and a singular value decay assumption on the operator $\mathcal{T}$.

**Assumption 6** (Singular Value Decay). *Let $p \geq 1$. There are constants $0 < c_1 \leq c_2 < \infty$ such that for all $i \geq 1$, $c_1 i^{-p} \leq \sigma_i \leq c_2 i^{-p}$.*

This assumption is standard in the nonparametric IV literature [8] and characterizes the difficulty of the inverse problem defined by the operator $\mathcal{T}$. A larger $p$ indicates faster decay and thus greater ill-posedness. Alternative ill-posedness characterizations can be considered, such as exponential decay [8].

**Assumption 7** (Source Condition). *Let $r > 0$. There is a constant $B < \infty$ such that*

$$\|h_0\|_r^2 := \left\|(\mathcal{T}^*\mathcal{T})^{-r/2} h_0\right\|_{L^2(X)}^2 = \sum_{i \geq 1} \frac{\langle h_0, u_i \rangle_{L^2(X)}^2}{\sigma_i^{2r}} \leq B.$$

This condition quantifies how "smooth" $h_0$ is relative to the spectral decomposition of $\mathcal{T}$, or equivalently, how quickly the alignment coefficients $\langle h_0, u_i \rangle_{L^2(X)}$ decay relative to the singular values. A larger $r$ implies $h_0$ is "smoother" or better aligned with the less ill-posed directions of $\mathcal{T}$. Note that under the "bad" scenario where the eigen-decay is fast ($p$ is large), a stronger alignment is necessary for $\|h_0\|_r^2$ to be finite for a large value of $r$. While in the "good" scenario with $p \sim 1$, $\|h_0\|_r^2 < +\infty$ is more likely to hold with a large value of $r$.

Under Assumption 7, we have

$$\|(\Pi_{\mathcal{X},d})_\perp h_0\|_{L_2(X)}^2 = \sum_{i>d} \langle h_0, u_i \rangle_{L^2(X)}^2 \leq \sigma_d^{2r} \sum_{i>d} \frac{|\langle h_0, u_i \rangle_{L^2(X)}|^2}{\sigma_i^{2r}} \leq \sigma_d^{2r} \|h_0\|_r^2.$$

Combined with Assumption 6, we obtain from Corollary 1 that:

$$\|\hat{h} - h_0\|_{L^2(X)} = \mathcal{O}_p\left(\|h_0\|_r d^{-rp} + \sqrt{\frac{d^{1+2p}}{n}}\right).$$

Balancing both terms in $d$ as a function of $n$, we obtain

$$\|\hat{h} - h_0\|_{L^2(X)} = \mathcal{O}_p\left(n^{-\frac{r}{2(r+1)+p^{-1}}}\right),$$

with

$$d(n) = n^{\frac{1}{2p(r+1)+1}}.$$

This bound is tight as a matching lower bound is obtained in Theorem 1-(i) [8]. From this tight rate, we see that both the ill-posedness ($p$) and the relative smoothness ($r$) fundamentally capture the effectiveness of 2SLS with spectral features, and their impacts are inherently intertwined.

- A low relative smoothness (small $r$) means $h_0$ has significant components aligned with directions corresponding to very small singular values, even after accounting for the singular value decay. This also directly slows the rate, effectively rendering the problem very hard or even impossible ($r \to 0_+$ makes the rate vacuous, representing our "ugly" scenario).
- A fast decay (large $p$) directly slows down the rate regardless of $r$. It means singular values drop quickly, making it fundamentally harder to resolve higher-frequency components of $h_0$. When $p$ is large we are either in the bad or in the ugly scenario. A large $p$ also slows down the rate indirectly by making it harder to achieve a high relative smoothness $r$.

**When working with learned features.** We see from Theorem 3 that when we learn the spectral features using contrastive learning, we incur an additional penalty related to $\hat{\varepsilon}_d/\sigma_d$ in the upper bound. This error shows that a faster eigendecay makes the 2SLS estimator more sensitive to errors in the learned features. Alignment of $h_0$ does not enter this factor, as learning the eigensubspaces of $\mathcal{T}$ is unrelated to $h_0$. Intuitively, features corresponding to small singular values can be empirically difficult to learn, even with a very large number of $(Z, X)$ samples.

## C Proofs

In this section we recall and prove the results of the main section.

### C.1 Proofs of Section 2

We refer the reader to [4] for background on measure-theoretic notions used throughout this section. We assume that $\mathcal{X}$ and $\mathcal{Z}$ are standard Borel spaces, *i.e.*, Borel subsets of complete separable metric spaces. This ensures the existence of regular conditional distributions and supports the disintegration of the joint law $\pi_{X,Z}$. Working directly with measures allows us to treat both continuous settings, where distributions admit densities with respect to the Lebesgue measure, and discrete or more general settings, where such densities may not exist.

To prove properties of the conditional expectation operator $\mathcal{T}$, it is helpful to express it in terms of a Markov kernel. We define a Markov kernel $p : \mathcal{Z} \times \mathcal{F}_{\mathcal{X}} \to [0,1]$ such that for any measurable set $A \in \mathcal{F}_{\mathcal{X}}$,

$$p(z, A) = \mathbb{P}[X \in A \mid Z = z],$$

meaning $p(z, \cdot)$ is a regular conditional distribution of $X$ given $Z = z$. Then for all $h \in L_2(X)$,

$$\mathcal{T}h(z) = \int_{\mathcal{X}} h(x)\, p(z, dx), \quad \pi_Z\text{-a.e.} \tag{11}$$

By the disintegration theorem, for all measurable sets $A \subset \mathcal{X}$ and $B \subset \mathcal{Z}$, the joint distribution admits the disintegration

$$\pi_{X,Z}(A \times B) = \int_B \left( \int_A p(z, dx) \right) d\pi_Z(z),$$

and therefore, the joint distribution $\pi_{X,Z}$ can be decomposed as

$$d\pi_{X,Z}(x, z) = p(z, dx)\, d\pi_Z(z), \tag{12}$$

where $p(z, dx)$ is a Markov kernel as defined above. This decomposition holds in general, even when $\pi_{X,Z}$ is not absolutely continuous with respect to $\pi_X \otimes \pi_Z$.

Under Assumption 2, the joint distribution $\pi_{X,Z}$ is absolutely continuous with respect to the product measure $\pi_X \otimes \pi_Z$, and thus admits a density

$$p(x, z) \doteq \frac{d\pi_{X,Z}}{d(\pi_X \otimes \pi_Z)}(x, z). \tag{13}$$

In order to prove Proposition 1, we use the following lemma.

**Lemma 1.** *Under Assumption 2, for $\pi_X \otimes \pi_Z$-almost every $(x, z)$,*

$$p(x, z) = \frac{dp(z, \cdot)}{d\pi_X}(x).$$

*Proof.* From Eq. (12), we have

$$d\pi_{X,Z}(x, z) = p(z, dx)\, d\pi_Z(z).$$

On the other hand, by Eq. (13),

$$d\pi_{X,Z}(x, z) = p(x, z)\, d\pi_X(x)\, d\pi_Z(z).$$

Comparing both expressions and using the uniqueness of the Radon–Nikodym derivative, we conclude that for $\pi_X \otimes \pi_Z$-almost every $(x, z)$,

$$p(z, dx) = p(x, z)\, d\pi_X(x),$$

*i.e.*, $p(z, \cdot) \ll \pi_X$, and

$$\frac{dp(z, \cdot)}{d\pi_X}(x) = p(x, z).$$

This concludes the proof. $\qquad\qquad\square$

By Lemma 1, under Assumption 2, for all $h \in L_2(X)$ and $\pi_Z$-a.e.,

$$\mathcal{T}h(z) = \int_{\mathcal{X}} h(x)\, p(z, dx) = \int_{\mathcal{X}} h(x) p(x, z) d\pi_X(x) = \langle h, p(\cdot, z) \rangle_{L_2(X)}. \tag{14}$$

We recall and prove Proposition 1. This is a classical result, see, *e.g.*, [26].

**Proposition 1.** *Under Assumption 2, $\mathcal{T}$ is a Hilbert–Schmidt operator and thus compact.*

*Proof.* Take $(e_i)_{i\geq 1}$ any orthonormal basis (ONB) of $L_2(X)$. Then, using the definition of the Hilbert–Schmidt norm:

$$
\begin{aligned}
\|\mathcal{T}\|_{\mathrm{HS}}^2 &= \sum_{i\geq 1} \|\mathcal{T}e_i\|_{L_2(Z)}^2 && \text{(definition of HS norm via ONB)} \\
&= \sum_{i\geq 1} \int_{\mathcal{Z}} |\mathcal{T}e_i(z)|^2 \, d\pi_Z(z) && \text{(expand $L_2(Z)$ norm)} \\
&= \sum_{i\geq 1} \int_{\mathcal{Z}} \langle e_i, p(\cdot,z)\rangle_{L_2(X)}^2 \, d\pi_Z(z) && \text{(by Eq. (14))} \\
&= \int_{\mathcal{Z}} \sum_{i\geq 1} \langle e_i, p(\cdot,z)\rangle_{L_2(X)}^2 \, d\pi_Z(z) && \text{(Fubini's theorem)} \\
&= \int_{\mathcal{Z}} \|p(\cdot,z)\|_{L_2(X)}^2 \, d\pi_Z(z) && \text{(Parseval's identity)} \\
&= \int_{\mathcal{X}\times\mathcal{Z}} p(x,z)^2 \, d\pi_X(x) \, d\pi_Z(z) && \text{(expand $L_2$ norm)} \\
&< +\infty && \text{(by Assumption 2, since $p \in L_2(\pi_X \otimes \pi_Z)$).}
\end{aligned}
$$

$\square$

Note that the proof shows in addition that $\|\mathcal{T}\|_{\mathrm{HS}} = \|p\|_{L_2(\pi_X \otimes \pi_Z)}$.

## C.2 Proofs of Section 3

**Proposition 4.** *Let $\mathcal{H}_{\varphi,d} = \mathrm{span}\{\varphi_1,\ldots,\varphi_d\}$, and let $\omega = \dim(\mathcal{H}_{\varphi,d})$. Then:*

   *i) $\tau_{\varphi,d} \geq \sigma_\omega^{-1}$.*
   *ii) If $\mathcal{H}_{\varphi,d} \subseteq \mathcal{V}_\iota = \mathrm{span}\{v_1,\ldots,v_\iota\}$, then $\tau_{\varphi,d} \leq \sigma_\iota^{-1}$.*
   *iii) For any $d \geq 1$, the minimal value $\tau_{\varphi,d} = \sigma_d^{-1}$ is achieved when $\mathcal{H}_{\varphi,d} = \mathcal{V}_d$.*

Proposition 4 is due to Lemma 1 of [3] and Proposition 2 corresponds to part (iii). We provide a full proof for completeness.

*Proof.* We first prove (i). Since $\dim(\mathcal{H}_{\varphi,d}) = \omega$, and the subspace $\mathcal{V}_{\omega-1}$ has dimension $\omega - 1$, there exists $\tilde{h} \in \mathcal{H}_{\varphi,d} \cap \mathcal{V}_{\omega-1}^\perp$ with $\|\tilde{h}\|_{L_2(X)} = 1$. Then,

$$
\begin{aligned}
\tau_{\varphi,d}^{-2} &= \inf_{\substack{h \in \mathcal{H}_{\varphi,d} \\ \|h\|_{L_2(X)}=1}} \|\mathcal{T}h\|_{L_2(Z)}^2 \\
&\leq \|\mathcal{T}\tilde{h}\|_{L_2(Z)}^2 \\
&\leq \sup_{\substack{h \in \mathcal{V}_{\omega-1}^\perp \\ \|h\|_{L_2(X)}=1}} \|\mathcal{T}h\|_{L_2(Z)}^2 \\
&= \sup_{\substack{h \in \mathcal{V}_{\omega-1}^\perp \\ \|h\|=1}} \langle \mathcal{T}^*\mathcal{T}h, h\rangle_{L_2(X)} = \sigma_\omega^2,
\end{aligned}
$$

where the last line follows from the min–max theorem for compact self-adjoint operators.

We now prove (ii). If $h \in \mathcal{H}_{\varphi,d} \subseteq \mathcal{V}_\iota$ and $\|h\|_{L_2(X)} = 1$, then

$$
h = \sum_{i=1}^{\iota} \langle h, v_i\rangle v_i,
$$

so by orthonormality of $(u_i)$,

$$\|\mathcal{T}h\|^2_{L_2(Z)} = \left\|\sum_{i=1}^{\iota} \sigma_i \langle h, v_i \rangle u_i \right\|^2_{L_2(Z)} = \sum_{i=1}^{\iota} \sigma_i^2 \langle h, v_i \rangle^2 \geq \sigma_\iota^2 \sum_{i=1}^{\iota} \langle h, v_i \rangle^2 = \sigma_\iota^2.$$

Taking the infimum over unit-norm $h \in \mathcal{H}_{\varphi,d}$ gives $\tau_{\varphi,d}^{-2} \geq \sigma_\iota^2$, *i.e.*, $\tau_{\varphi,d} \leq \sigma_\iota^{-1}$.

Finally, (iii) follows by taking $\mathcal{H}_{\varphi,d} = \mathcal{V}_d$, so that $\omega = d = \iota$, and both bounds in (i) and (ii) match. $\qquad\square$

We recall and prove Corollary 1:

**Corollary 1** (Sieve 2SLS with spectral features). *Let $\hat{h}$ be the 2SLS estimator from Eq. (5) using spectral features. Let Assumptions 1, 2 and 4 hold and $\sigma_d^{-1}\zeta_{\varphi,\psi,d}\sqrt{(\log d)/n} = o(1)$. Then:*

$$\left\|\hat{h} - h_0\right\|_{L^2(X)} = O_p\left(\|(\Pi_{\mathcal{X},d})_\perp h_0\|_{L^2(X)} + \sqrt{\frac{d}{n\sigma_d^2}}\right).$$

*Proof.* Let us verify that Assumption 3 holds under Definition 2, *i.e.*, when $\mathcal{H}_{\varphi,d} = \mathcal{V}_d$ and $\mathcal{H}_{\psi,d} = \mathcal{U}_d$.

(i) Since the image of $\mathcal{V}_d$ under $\mathcal{T}$ lies in $\mathcal{U}_d$, we have

$$(\Pi_{\psi,d})_\perp \mathcal{T}h = 0 \quad \text{for all } h \in \mathcal{H}_{\varphi,d}.$$

Hence,

$$\sup_{\substack{h\in\mathcal{H}_{\varphi,d} \\ \|h\|_{L_2}=1}} \|(\Pi_{\psi,d})_\perp \mathcal{T}h\|_{L^2(Z)} = 0 = o\left(\tau_{\varphi,d}^{-1}\right),$$

and condition (i) is satisfied.

(ii) Let $h_0 \in L_2(X)$, and consider its projection onto the orthogonal complement of $\mathcal{H}_{\varphi,d}$. Since $\mathcal{T}$ is compact with singular value decomposition $\mathcal{T} = \sum_{i=1}^{\infty} \sigma_i u_i \otimes v_i$ under Assumption 2, we have

$$\|\mathcal{T}(\Pi_{\varphi,d})_\perp h_0\|^2_{L_2(Z)} = \sum_{i>d} \sigma_i^2 \langle h_0, v_i \rangle^2 \leq \sigma_d^2 \sum_{i>d} \langle h_0, v_i \rangle^2 = \sigma_d^2 \|(\Pi_{\varphi,d})_\perp h_0\|^2_{L_2(X)}.$$

This implies, using that $\tau_{\varphi,d} = \sigma_d^{-1}$ by Proposition 2,

$$\|\mathcal{T}(\Pi_{\varphi,d})_\perp h_0\|_{L_2(Z)} \leq \sigma_d \|(\Pi_{\varphi,d})_\perp h_0\|_{L_2(X)} = \tau_{\varphi,d}^{-1}\|(\Pi_{\varphi,d})_\perp h_0\|_{L_2(X)},$$

so Assumption 3-(ii) holds with constant $C = 1$.

Since both parts of Assumption 3 hold, and $\tau_{\varphi,d} = \sigma_d^{-1}$ by Proposition 2, we may apply Theorem 1, which yields the desired result. $\qquad\square$

### C.3 Proofs of Section 4

The following result shows that Eq. (9) holds if and only if $\text{rank}(\mathcal{T}) \leq d$.

**Proposition 5.** *Let $\varphi : \mathcal{X} \to \mathbb{R}^d$, $\psi : \mathcal{Z} \to \mathbb{R}^d$ with components $\varphi_i \in L_2(X)$, $\psi_i \in L_2(Z)$. Then*

$$p(x,z) = \varphi(x)^\intercal \psi(z) \quad \text{if and only if} \quad \mathcal{T} = \sum_{i=1}^{d} \psi_i \otimes \varphi_i.$$

*Proof.* $\Rightarrow$ Assume $p(x,z) = \varphi(x)^\intercal \psi(z)$. Then, by Eq. (11), for all $h \in L_2(X)$,

$$\mathcal{T}h(z) = \int h(x)p(z,dx) = \int h(x)p(x,z)\,d\pi_X(x) \qquad \text{(by Lemma 1)}$$

$$= \int h(x)\varphi(x)^\intercal \psi(z)\,d\pi_X(x) = \psi(z)^\intercal \int h(x)\varphi(x)\,d\pi_X(x)$$

$$= \sum_{i=1}^{d} \psi_i(z)\langle h, \varphi_i \rangle_{L_2(X)} = \left(\sum_{i=1}^{d} \psi_i \otimes \varphi_i\right)(h)(z),$$

so $\mathcal{T} = \sum_{i=1}^{d} \psi_i \otimes \varphi_i$.

$\Leftarrow$ Now assume $\mathcal{T} = \sum_{i=1}^{d} \psi_i \otimes \varphi_i$. Let $f \in L_2(\pi_X \otimes \pi_Z)$. We want to show:

$$\int f(x,z)p(x,z)\, d(\pi_X \otimes \pi_Z)(x,z) = \int f(x,z)\varphi(x)^{\mathsf{T}}\psi(z)\, d(\pi_X \otimes \pi_Z)(x,z).$$

Starting with the right-hand side:

$$\int f(x,z)\, \varphi(x)^{\mathsf{T}}\psi(z)\, d\pi_X(x)d\pi_Z(z) = \sum_{i=1}^{d} \int f(x,z)\varphi_i(x)\psi_i(z)\, d\pi_X(x)d\pi_Z(z)$$

$$= \sum_{i=1}^{d} \int_{\mathcal{Z}} \psi_i(z) \left( \int_{\mathcal{X}} f(x,z)\varphi_i(x)\, d\pi_X(x) \right) d\pi_Z(z)$$

$$= \int_{\mathcal{Z}} \left( \sum_{i=1}^{d} \psi_i(z)\langle f(\cdot,z), \varphi_i \rangle \right) d\pi_Z(z)$$

$$= \int_{\mathcal{Z}} \left[ \mathcal{T} f(\cdot,z) \right](z)\, d\pi_Z(z) \quad (\text{since } \mathcal{T} = \sum \psi_i \otimes \varphi_i)$$

$$= \int_{\mathcal{Z}} \left( \int_{\mathcal{X}} f(x,z)\, p(z,dx) \right) d\pi_Z(z) \quad (\text{by Eq. (11)})$$

$$= \int_{\mathcal{X} \times \mathcal{Z}} f(x,z)p(x,z)\, d\pi_X(x)d\pi_Z(z) \quad (\text{by Lemma 1}).$$

Since both integrals agree for all $f \in L_2(\pi_X \otimes \pi_Z)$, and $p(x,z)$ is the Radon–Nikodym derivative of $\pi_{X,Z}$ with respect to $\pi_X \otimes \pi_Z$, it follows by uniqueness that

$$p(x,z) = \varphi(x)^{\mathsf{T}}\psi(z) \quad \text{for } (\pi_X \otimes \pi_Z)\text{-almost every } (x,z).$$

$\square$

In order to prove Theorem 3 we prove intermediate results.

**Proposition 6.** *Assumption 5 holds if and only if* $\mathrm{rank}(\hat{\mathcal{T}}_d) = d$.

*Proof.* Assume that Assumption 5 holds, *i.e.*, $\{\hat{\varphi}_1, \ldots, \hat{\varphi}_d\}$ and $\{\hat{\psi}_1, \ldots, \hat{\psi}_d\}$ are linearly independent. We show that $\mathrm{rank}(\hat{\mathcal{T}}_d) = d$.

Since $\hat{\mathcal{T}}_d h = \sum_{i=1}^{d} \langle h, \hat{\varphi}_i \rangle \hat{\psi}_i$, we can write $\hat{\mathcal{T}}_d = A \circ \Phi$, where:

- $\Phi : L_2(X) \to \mathbb{R}^d$, defined by $h \mapsto (\langle h, \hat{\varphi}_1 \rangle, \ldots, \langle h, \hat{\varphi}_d \rangle)$,
- $A : \mathbb{R}^d \to L_2(Z)$, defined by $a \mapsto \sum_{i=1}^{d} a_i \hat{\psi}_i$.

Under Assumption 5, the family $\{\hat{\varphi}_i\}$ is linearly independent, so $\Phi$ is surjective. Similarly, the family $\{\hat{\psi}_i\}$ is linearly independent, so $A$ is injective. Therefore, the composition $A \circ \Phi$ has image of dimension

$$\mathrm{rank}(\hat{\mathcal{T}}_d) = \dim(\mathcal{R}(A \circ \Phi)) = \dim(\mathcal{R}(A)) = d,$$

where the second equality follows from the surjectivity of $\Phi$, and the third from the injectivity of $A$.

Conversely, suppose $\mathrm{rank}(\hat{\mathcal{T}}_d) = d$. Then $\hat{\mathcal{T}}_d$ has image of dimension $d$, which implies that the $\hat{\psi}_i$ must be linearly independent. The same reasoning applied to $\mathcal{T}^*$ shows that $\{\hat{\varphi}_1, \ldots, \hat{\varphi}_d\}$ is linearly independent.

$\square$

Under Assumption 5, we can therefore write $\hat{\mathcal{T}}_d$ in the following SVD form:

$$\hat{\mathcal{T}}_d = \sum_{i=1}^{d} \hat{\sigma}_i \hat{u}_i \otimes \hat{v}_i, \tag{15}$$

where $\hat{\sigma}_1 \geq \hat{\sigma}_2 \geq \cdots \geq \hat{\sigma}_d > 0$ are the singular values of $\hat{\mathcal{T}}$ and $\hat{u}_i$ and $\hat{v}_i$ are the left and right singular functions of $\hat{\mathcal{T}}$.

**Proposition 7.** *Under Assumption 5,* $\mathcal{H}_{\hat{\varphi},d} = \operatorname{span}\{\hat{v}_1, \ldots, \hat{v}_d\}$ *and* $\mathcal{H}_{\hat{\psi},d} = \operatorname{span}\{\hat{u}_1, \ldots, \hat{u}_d\}$.

*Proof.* We have by definition of the SVD that
$$\mathcal{N}(\hat{\mathcal{T}}_d)^\perp = \operatorname{span}\{\hat{v}_1, \ldots, \hat{v}_d\}.$$
To show that $\mathcal{H}_{\hat{\varphi},d} = \operatorname{span}\{\hat{v}_1, \ldots, \hat{v}_d\}$, it suffices to prove that
$$\mathcal{N}(\hat{\mathcal{T}}_d) = \operatorname{span}\{\hat{\varphi}_1, \ldots, \hat{\varphi}_d\}^\perp.$$
Let $h \in L_2(X)$. Then:
$$h \in \mathcal{N}(\hat{\mathcal{T}}_d) \iff \hat{\mathcal{T}}_d h = 0 = \sum_{i=1}^d \langle h, \hat{\varphi}_i \rangle \hat{\psi}_i$$

$$\iff \sum_{i=1}^d \langle h, \hat{\varphi}_i \rangle \hat{\psi}_i = 0$$

$$\iff \langle h, \hat{\varphi}_i \rangle = 0 \quad \forall i \in [d],$$

where the last equivalence follows from the linear independence of $\{\hat{\psi}_i\}_{i=1}^d$.

Therefore, $\mathcal{N}(\hat{\mathcal{T}}_d) = \operatorname{span}\{\hat{\varphi}_1, \ldots, \hat{\varphi}_d\}^\perp$, which implies
$$\mathcal{H}_{\hat{\varphi},d} = \mathcal{N}(\hat{\mathcal{T}}_d)^\perp = \operatorname{span}\{\hat{v}_1, \ldots, \hat{v}_d\}.$$

The proof for $\mathcal{H}_{\hat{\psi},d} = \operatorname{span}\{\hat{u}_1, \ldots, \hat{u}_d\}$ is analogous, using the adjoint $\hat{\mathcal{T}}_d^*$. $\qquad\square$

We now recall and prove Theorem 3:

**Theorem 3.** *Let Assumptions 1, 2, 4 and 5 hold and let* $\hat{\varepsilon}_d \doteq \left\|\hat{\mathcal{T}}_d - \mathcal{T}_d\right\|_{\text{op}}$ *be such that* $\hat{\varepsilon}_d < (1 - 1/\sqrt{2})\sigma_d$.

i) $\sigma_d^{-1} \leq \tau_{\hat{\varphi},d} \leq (\sigma_d - 2\hat{\varepsilon}_d)^{-1}$;

ii) *Let* $\hat{h}$ *be the 2SLS estimator from Eq. (5), using features* $\hat{\varphi} : \mathcal{X} \to \mathbb{R}^d$ *and* $\hat{\psi} : \mathcal{Z} \to \mathbb{R}^d$. *Suppose* $(\sigma_d - 2\hat{\varepsilon}_d)^{-1} \zeta_{\hat{\varphi},\hat{\psi},d} \sqrt{(\log d)/n} = o(1)$ *and* $\hat{\varepsilon}_d = o_d(\sigma_d^2)$, *then*

$$\left\|\hat{h} - h_0\right\|_{L^2(X)} = O_p\left(\|(\Pi_{\mathcal{X},d})_\perp h_0\|_{L_2(X)} + \frac{\sqrt{2}\hat{\varepsilon}_d \|h_0\|_{L_2(X)}}{\sigma_d} + \frac{1}{\sigma_d - 2\hat{\varepsilon}_d}\sqrt{\frac{d}{n}}\right).$$

*Proof.* We first show that $\sigma_d^{-1} \leq \tau_{\hat{\varphi},d} \leq (\sigma_d - 2\hat{\varepsilon}_d)^{-1}$. First note that if $h \in \mathcal{H}_{\hat{\varphi},d}$, then by Proposition 7, $h \in \mathcal{V}_d$. Using the SVD of $\hat{\mathcal{T}}_d$ from Eq. (15), and writing $h = \sum_{i=1}^d \langle h, \hat{v}_i \rangle \hat{v}_i$, we obtain:

$$\hat{\mathcal{T}}_d h = \sum_{i=1}^d \hat{\sigma}_i \langle h, \hat{v}_i \rangle \hat{u}_i, \quad \text{so} \quad \|\hat{\mathcal{T}}_d h\|_{L_2(Z)}^2 = \sum_{i=1}^d \hat{\sigma}_i^2 \langle h, \hat{v}_i \rangle^2 \geq \hat{\sigma}_d^2 \|h\|_{L_2(X)}^2.$$

Next, observe that for all $h \in L_2(X)$,
$$\|\mathcal{T}h\|_{L_2(Z)} \geq \|\mathcal{T}_d h\|_{L_2(Z)}.$$

By the reverse triangle inequality, the bound $\left\|\hat{\mathcal{T}}_d - \mathcal{T}_d\right\|_{\text{op}} \leq \hat{\varepsilon}_d$, and Weyl's inequality for singular values, we have, for all $h \in L_2(X)$, with $\|h\|_{L_2(X)} = 1$:
$$\|\mathcal{T}h\|_{L_2(Z)} \geq \|\mathcal{T}_d h\|_{L_2(Z)}$$
$$\geq \|\hat{\mathcal{T}}_d h\|_{L_2(Z)} - \left\|\hat{\mathcal{T}}_d - \mathcal{T}_d\right\|_{\text{op}} \cdot \|h\|_{L_2(X)}$$
$$\geq (\hat{\sigma}_d - \hat{\varepsilon}_d)\|h\|_{L_2(X)}$$
$$= \hat{\sigma}_d - \hat{\varepsilon}_d$$
$$\geq \sigma_d - 2\hat{\varepsilon}_d.$$

This shows that
$$\tau_{\hat{\varphi},d}^{-1} \geq \sigma_d - 2\hat{\varepsilon}_d \quad \Rightarrow \quad \tau_{\hat{\varphi},d} \leq (\sigma_d - 2\hat{\varepsilon}_d)^{-1},$$

which holds under the assumption $\hat{\varepsilon}_d < (1 - 1/\sqrt{2})\sigma_d < \sigma_d/2$. On the other hand, since $\dim(\mathcal{H}_{\hat{\varphi},d}) = d$ by Assumption 5, Proposition 4-(i) implies

$$\sigma_d \quad \Rightarrow \quad \tau_{\hat{\varphi},d} \geq \sigma_d^{-1}.$$

Next, we show how to bound the sieve approximation error, write:

$$
\begin{aligned}
\left\|(\Pi_{\hat{\varphi},d})_{\perp} h_0\right\|_{L_2(X)} &= \left\|(\Pi_{\hat{\varphi},d})_{\perp} \left((\Pi_{\mathcal{X},d})_{\perp} h_0 + \Pi_{\mathcal{X},d} h_0\right)\right\|_{L_2(X)} \\
&\leq \left\|(\Pi_{\mathcal{X},d})_{\perp} h_0\right\|_{L_2(X)} + \left\|(\Pi_{\hat{\varphi},d})_{\perp} \Pi_{\mathcal{X},d} h_0\right\|_{L_2(X)} \\
&\leq \left\|(\Pi_{\mathcal{X},d})_{\perp} h_0\right\|_{L_2(X)} + \left\|(\Pi_{\hat{\varphi},d})_{\perp} \Pi_{\mathcal{X},d}\right\|_{\mathrm{op}} \cdot \|h_0\|_{L_2(X)}.
\end{aligned}
$$

By Wedin's sin–$\Theta$ theorem (Theorem 2.9 and Eq. (2.26a) in [9]), and under the assumption $\hat{\varepsilon}_d < (1 - 1/\sqrt{2})\sigma_d$, we have:

$$\left\|(\Pi_{\hat{\varphi},d})_{\perp} \Pi_{\mathcal{X},d}\right\|_{\mathrm{op}} \leq \frac{\sqrt{2}\hat{\varepsilon}_d}{\sigma_d}.$$

Therefore,

$$\left\|(\Pi_{\hat{\varphi},d})_{\perp} h_0\right\|_{L_2(X)} \leq \left\|(\Pi_{\mathcal{X},d})_{\perp} h_0\right\|_{L_2(X)} + \frac{\sqrt{2}\hat{\varepsilon}_d \|h_0\|_{L_2(X)}}{\sigma_d},$$

To conclude we now check Assumption 3. Let us introduce

$$s_{\hat{\psi},\hat{\varphi},d}^{-1} = \sup_{h \in \mathcal{H}_{\hat{\varphi},d}, \, h \neq 0} \frac{\|h\|_{L_2(X)}}{\|\Pi_{\hat{\psi},d} \mathcal{T} h\|_{L_2(Z)}} \geq \tau_{\hat{\varphi},d}.$$

The unique use of Assumption 3-i) in the proof of Theorem B.1 [6] is to prove that $s_{\hat{\psi},\hat{\varphi},d}^{-1} = O(\tau_{\hat{\varphi},d})$ as $d \to +\infty$, which we now directly prove. First, note that for any $h \in L_2(X)$,

$$
\begin{aligned}
\left\|\mathcal{T}\Pi_{\hat{\varphi},d} h\right\|_{L_2(Z)} &= \left\|\mathcal{T}\left(\Pi_{\hat{\varphi},d} + \Pi_{\mathcal{X},d} - \Pi_{\mathcal{X},d}\right) h\right\|_{L_2(Z)} \\
&\leq \|\mathcal{T}_d h\|_{L_2(Z)} + \|\Pi_{\hat{\varphi},d} - \Pi_{\mathcal{X},d}\|_{\mathrm{op}} \|h\|_{L_2(X)} \\
&\leq \|\mathcal{T}_d h\|_{L_2(Z)} + \frac{\sqrt{2}\hat{\varepsilon}_d \|h\|_{L_2(X)}}{\sigma_d},
\end{aligned}
$$

where we used $\|\mathcal{T}\|_{\mathrm{op}} \leq 1$, the definition of $\mathcal{T}_d$ and Wedin's sin–$\Theta$ theorem. On the other hand,

$$
\begin{aligned}
\left\|\Pi_{\hat{\psi},d} \mathcal{T} \Pi_{\hat{\varphi},d} h\right\|_{L_2(Z)} &\geq \left\|\Pi_{\hat{\psi},d} \mathcal{T} \Pi_{\varphi,d} h\right\|_{L_2(Z)} - \left\|\Pi_{\hat{\psi},d} \mathcal{T}\left(\Pi_{\hat{\varphi},d} - \Pi_{\mathcal{X},d}\right) h\right\|_{L_2(Z)} \\
&\geq \left\|\Pi_{\psi,d} \mathcal{T} \Pi_{\varphi,d} h\right\|_{L_2(Z)} - \left\|(\Pi_{\psi,d} - \Pi_{\hat{\psi},d}) \mathcal{T} \Pi_{\varphi,d} h\right\|_{L_2(Z)} - \frac{\sqrt{2}\hat{\varepsilon}_d \|h\|_{L_2(X)}}{\sigma_d} \\
&\geq \|\mathcal{T}_d h\|_{L_2(Z)} - \frac{2\sqrt{2}\hat{\varepsilon}_d \|h\|_{L_2(X)}}{\sigma_d}.
\end{aligned}
$$

We therefore obtain that,

$$s_{\hat{\psi},\hat{\varphi},d}^{-1} \leq \sup_{h \in \mathcal{H}_{\hat{\varphi},d}, \, h \neq 0} \frac{1}{\frac{\|\mathcal{T}\Pi_{\hat{\varphi},d} h\|_{L_2(Z)}}{\|h\|_{L_2(X)}} - \frac{3\sqrt{2}\hat{\varepsilon}_d}{\sigma_d}} \leq \tau_{\hat{\varphi},d} \times \frac{1}{1 - \frac{3\sqrt{2}\hat{\varepsilon}_d \tau_{\hat{\varphi},d}}{\sigma_d}},$$

where the inequality is valid for $d$ large enough as long as $3\sqrt{2}\hat{\varepsilon}_d \tau_{\hat{\varphi},d} \sigma_d^{-1} = o(1)$. Re-using that $\tau_{\hat{\varphi},d} \leq (\sigma_d - 2\hat{\varepsilon}_d)^{-1}$, a sufficient condition is

$$3\sqrt{2} \frac{\hat{\varepsilon}_d}{\sigma_d} \frac{1}{\sigma_d - 2\hat{\varepsilon}_d} = o(1),$$

which is satisfied if $\hat{\varepsilon}_d = o(\sigma_d^2)$. We therefore conclude that if $\hat{\varepsilon}_d = o(\sigma_d^2)$ then $s_{\hat{\psi},\hat{\varphi},d}^{-1} = O(\tau_{\hat{\varphi},d})$. Finally, we check Assumption 3-ii). First note that by definition of $\Pi_{\hat{\varphi},d}$ and $\hat{\mathcal{T}}_d$, we have

$\hat{\mathcal{T}}_d(\Pi_{\hat{\varphi},d})_\perp = 0$, therefore,

$$
\begin{aligned}
\tau_{\hat{\varphi},d} \left\| \mathcal{T}(\Pi_{\hat{\varphi},d})_\perp h_0 \right\| = \tau_{\hat{\varphi},d} &\left\| (\mathcal{T} - \hat{\mathcal{T}}_d)(\Pi_{\hat{\varphi},d})_\perp h_0 \right\| \\
&\leq \tau_{\hat{\varphi},d} \left( \left\| (\mathcal{T} - \mathcal{T}_d)(\Pi_{\hat{\varphi},d})_\perp h_0 \right\| + \left\| (\mathcal{T}_d - \hat{\mathcal{T}}_d)(\Pi_{\hat{\varphi},d})_\perp h_0 \right\| \right) \\
&\leq \tau_{\hat{\varphi},d}(\sigma_{d+1} + \hat{\varepsilon}_d) \left\| (\Pi_{\hat{\varphi},d})_\perp h_0 \right\| \\
&\leq \frac{\sigma_{d+1} + \hat{\varepsilon}_d}{\sigma_d - 2\hat{\varepsilon}_d} \left\| (\Pi_{\hat{\varphi},d})_\perp h_0 \right\|,
\end{aligned}
$$

We conclude using that $\hat{\varepsilon}_d = o(\sigma_d^2)$ implies $\hat{\varepsilon}_d = o(\sigma_d)$ which implies $(\sigma_{d+1} + \hat{\varepsilon}_d)(\sigma_d - 2\hat{\varepsilon}_d) = O(1)$. $\qquad\square$

We now prove the equivalence between Eq. (6) and Eq. (7). While the proof strategy follows that of Kostic et al. [25, Theorem 1], we include our own version for completeness, as the parametrization of the learned operator differs. Specifically, [25] directly approximate the truncated operator $\mathcal{T}_d$ using a singular value decomposition of the form $\mathbb{1}_\mathcal{Z} \otimes \mathbb{1}_\mathcal{X} + \sum_{i=2}^d \hat{\sigma}_i \hat{\psi}_i \otimes \hat{\varphi}_i$, where the singular values $\hat{\sigma}_i$ are learned explicitly via a separate network. In contrast, our approach learns only the feature maps $(\varphi_i, \psi_i)$ and defines the approximation $\mathcal{T}_d(\hat{\varphi}, \hat{\psi}) = \sum_{i=1}^d \hat{\psi}_i \otimes \hat{\varphi}_i$. As such, we provide a short self-contained derivation adapted to our setting for clarity. Recall that the Eckart–Young–Mirsky formulation of the objective is defined as

$$
\mathcal{L}_d(\varphi, \psi) = \left\| \mathcal{T}_d(\varphi, \psi) - \mathcal{T} \right\|_{\text{HS}}^2 = \left\| \mathcal{T}_d(\varphi, \psi) \right\|_{\text{HS}}^2 - 2\langle \mathcal{T}_d(\varphi, \psi), \mathcal{T} \rangle_{\text{HS}} + \left\| \mathcal{T}_d(\varphi, \psi) \right\|_{\text{HS}}^2.
$$

**Proposition 8.** *It holds that,*

$$
\left\| \mathcal{T}_d(\varphi, \psi) \right\|_{\text{HS}}^2 - 2\langle \mathcal{T}_d(\varphi, \psi), \mathcal{T} \rangle_{\text{HS}} = \mathbb{E}_X \mathbb{E}_Z \left[ (\varphi(X)^\intercal \psi(Z))^2 \right] - 2\mathbb{E}_{X,Z} \left[ \varphi(X)^\intercal \psi(Z) \right].
$$

*Proof.* Let us introduce

$$
\Phi : \mathbb{R}^d \to L_2(X), \quad c \mapsto \sum_{i=1}^d c_i \varphi_i, \qquad \Psi : \mathbb{R}^d \to L_2(Z), \quad c \mapsto \sum_{i=1}^d c_i \psi_i,
$$

such that $\mathcal{T}_d(\varphi, \psi) = \Psi\Phi^*$, $\Psi^*\Psi = \mathbb{E}[\psi(Z)\psi(Z)^\intercal]$ and $\Phi^*\Phi = \mathbb{E}[\varphi(X)\varphi(X)^\intercal]$. On one hand, we have:

$$
\begin{aligned}
\left\| \mathcal{T}_d(\varphi, \psi) \right\|_{\text{HS}}^2 &= \left\| \Psi\Phi^* \right\|_{\text{HS}}^2 \\
&= \text{Tr}\left( \Psi^*\Psi\Phi^*\Phi \right) \\
&= \text{Tr}\left( \mathbb{E}[\psi(Z)\psi(Z)^\intercal]\mathbb{E}[\varphi(X)\varphi(X)^\intercal] \right) \\
&= \mathbb{E}_X \mathbb{E}_Z \left[ \text{Tr}\left( \psi(Z)\psi(Z)^\intercal\varphi(X)\varphi(X)^\intercal \right) \right] \\
&= \mathbb{E}_X \mathbb{E}_Z \left[ (\psi(Z)^\intercal\varphi(X))^2 \right].
\end{aligned}
$$

On the other hand, we have:

$$
\begin{aligned}
\langle \mathcal{T}_d(\varphi, \psi), \mathcal{T} \rangle_{\text{HS}} &= \text{Tr}\left( \Psi^*\mathcal{T}\Phi \right) \\
&= \text{Tr}\left( \mathbb{E}\left[ \psi(Z)\mathbb{E}\left[ \varphi(X) \mid Z \right]^\intercal \right] \right) \\
&= \mathbb{E}_{XZ} \left[ \text{Tr}\left( \psi(Z)\varphi(X)^\intercal \right) \right] \\
&= \mathbb{E}_{XZ} \left[ \psi(Z)^\intercal\varphi(X) \right],
\end{aligned}
$$

which conclude the proof. $\qquad\square$

As noted in [25], the above result does not require $\mathcal{T}$ to be Hilbert-Schmidt, or even compact. Indeed as $\mathcal{T}_d(\varphi, \psi)$ is a finite rank operator, $\langle \mathcal{T}_d(\varphi, \psi), \mathcal{T} \rangle_{\text{HS}}$ is always well defined.

# D  Extended Related Work

We now discuss the various ideas that have been proposed to solve NPIV problems. Extensive benchmarking for these methods has already been conducted in prior work; refer, for instance, to [36].

As mentioned in Section 1, a central challenge in NPIV estimation is that it constitutes an ill-posed inverse problem. The literature has evolved along several methodological lines to address it. Two broad classes of estimation strategies have become particularly prominent:

- **Two-Stage Least-Squares (2SLS).** This classical approach and its various generalizations involves a sequential estimation procedure. Typically, the first stage estimates the conditional expectation of the endogenous variable (or its features) given the instrument, and the second stage uses these predictions to estimate the structural function.
- **Saddle-point optimization.** These methods, often rooted in a generalized method of moments (GMM) framework or duality principles, reformulate the NPIV estimation as finding an equilibrium in a min-max game. This often involves optimizing an objective function over a hypothesis space for the structural function and a test function space for moment conditions.

These two methodologies developed with distinct focuses. The 2SLS methods, with roots in classical econometrics [32, 11], have progressively incorporated more sophisticated nonparametric techniques. Saddle-point methods, on the other hand, have gained traction with the rise of machine learning, offering powerful tools for handling complex optimization problems arising from conditional moment restrictions [13, 29, 2].

**2SLS approaches.** Early and influential nonparametric extensions of 2SLS employed sieve or series estimators. These methods approximate the unknown functions using a finite linear combination of basis functions, such as polynomials, splines, or Fourier series. The number of basis functions, or the "sieve dimension", is allowed to increase with the sample size, enabling consistent estimation of the nonparametric functions.

A seminal contribution, by [32], provided identification results and a consistent nonparametric estimator for conditional moment restrictions. Their proposed NPIV estimator is an analog of 2SLS, where the first stage involves nonparametric estimation of conditional means of basis functions of $X$ given $Z$, and the second stage uses a series approximation for the structural function based on these first-stage predictions. Regularization is achieved by controlling the number of terms in the series approximation.

Building on similar principles, [18] proposed nonparametric methods based on both kernel[6] techniques and orthogonal series for estimating regression functions with instrumental variables. For their orthogonal series estimator, they transformed the marginal distributions of $X$ and $Z$ to be uniform and used Fourier expansions. The estimated Fourier coefficients of the structural function were obtained via a regularized regression involving estimated coefficients from the first stage. They were among the first to derive optimal convergence rates for this class of problems, explicitly linking these rates to the "difficulty" of the ill-posed inverse problem, which is characterized by the eigenvalues of the underlying integral operator.

[11] proposed an estimation procedure based on Tikhonov regularization. This involves regularizing the inverse of the integral operator $\mathcal{T}$ (or its empirical counterpart) to stabilize the solution. Specifically, denoting $\hat{\mathcal{T}}$ and $\hat{r}_0$ empirical estimates (in their case, computed with kernel density estimators), the Tikhonov regularized solution is of the form $h_\alpha = \left( \hat{\mathcal{T}}^\star \hat{\mathcal{T}} + \alpha I \right)^{-1} \hat{\mathcal{T}}^\star \hat{r}_0$, where $\alpha > 0$. They presented asymptotic properties of their estimator, including consistency and convergence rates, which depend on the smoothness of the structural function and the degree of ill-posedness of $\mathcal{T}$.

[35] introduced Kernel Instrumental Variable Regression (KIV), a direct nonparametric generalization of 2SLS. KIV models the relationships between the different variables as nonlinear functions in reproducing kernel Hilbert spaces (RKHSs). In stage 1, it learns a conditional mean embedding $\mu(z) = \mathbb{E}\left[\varphi(X) \mid Z = z\right]$ where $\varphi(X)$ represents features of $X$ mapped into an RKHS $\mathcal{H}_X$. This learning is framed as a vector-valued kernel ridge regression, effectively estimating the conditional

---

[6]"Kernel" is meant here in the sense of density estimation, and not reproducing kernel: see [18, eq. (2.4)].

expectation operator $E : \mathcal{H}_X \to \mathcal{H}_Z$. In stage 2, a scalar-valued kernel ridge regression of the outcome $Y$ on the estimated means $\hat{\mu}(Z)$ is performed to estimate the structural function. The authors prove the consistency of KIV in the projected norm under mild conditions and derive conditions under which KIV achieves minimax optimal convergence rates. The analysis was later improved by [31], who established convergence in $L_2$ norm rather than the projected norm.

Another related approach, "fast IV" (FIV), was proposed in [39]. FIV studies nonlinear IV with high-dimensional instruments and proposes a two-stage pipeline that keeps the outcome model in a fixed RKHS/GP space but learns the first-stage kernel from a black-box adaptive regressor (*e.g.*, a neural network) distilled into a compact kernel basis for estimating the conditional expectation operator. The resulting estimator plugs this learned kernel into standard kernelized IV schemes, achieving rates that adapt to the dimensionality of informative instrument features. Their work focuses on learning a kernel basis for the instrument. In contrast, our method learns paired spectral features for both the instrument and the treatment to approximate the top singular subspaces of the conditional expectation operator.

**Saddle-point approaches.**    These arise from reformulating the conditional moment restrictions that define the NPIV problem (Eq. 1), frequently conceptualized as a zero-sum game. One player selects a function $h$ from a hypothesis space $\mathcal{H}$ to minimize the objective. The other player, the "adversary" or "witness", selects a test function $g$ from a test function space $\mathcal{G}$ to maximize the objective. The adversary's role is to select the function that maximizes the violation of the moment condition by the current choice of $h$. A notable advantage of this formulation is that it can allow bypassing the direct estimation of conditional expectations. Saddle-point objectives in NPIV regression can be derived in different ways.

One can frame the NPIV problem as finding a solution $h_0$ that minimizes a certain criterion (*e.g.*, a norm) subject to satisfying the moment condition $\mathcal{T}h_0 = r_0$. The Lagrangian of this constrained problem then leads to a minimax objective. [2] target the least-norm solution

$$h_0 = \arg\min_{h \in L_2(X)} \frac{1}{2} \|h\|_{L_2(X)}^2 \quad \text{subject to} \quad \mathcal{T}h = r_0 \,.$$

The corresponding Lagrangian is given by $L(f,g) = \frac{1}{2} \|h\|_{L_2(X)}^2 + \langle r_0 - \mathcal{T}h, g \rangle_{L_2(Z)}$ for $g \in L_2(Z)$. Their method achieves strong $L_2$ error rates under a source condition and realizability assumptions. Notably, their approach does not require the often-problematic closedness condition or uniqueness of the IV solution. [29] consider a similar approach, focusing on the setting where the function classes are formed by neural networks, but use different techniques to analyze their method (online learning and neural tangent kernel theory).

Another approach to deriving a saddle-point problem involves using Fenchel duality to transform a squared error loss involving conditional expectations. This is the approach taken by [36], which we study in this paper.

A third method is to directly use the unconditional moment formulation $\mathbb{E}\left[(Y - h(X)) g(Z)\right] = 0$. [13] define their criterion as the maximum moment deviation,

$$h_0 = \arg\inf_{h \in \mathcal{H}} \sup_{g \in \mathcal{G}} \mathbb{E}\left[(Y - h(X)) g(Z)\right] = \Psi(h, g) \,,$$

and define the estimator $\hat{h} = \arg\min_h \sup_g \mathbb{E}_n\left[(Y - h(X)) g(Z)\right] + \mu R_1(h) - \lambda R_2(g)$, where $R_1$, $R_2$ are regularizers. Their key theoretical result is that the statistical estimation rate, in terms of projected mean squared error $\|\mathcal{T}(\hat{h} - h_0)\|_2$, scales with the critical radii of the hypothesis space $\mathcal{H}$ and the test function space $\mathcal{G}$. This holds under some closedness assumption, namely that $\mathbb{E}\left[h(X) - h'(X) \mid Z\right] \in \mathcal{G}$ for any $h, h' \in \mathcal{H}$. We note that the method introduced by [43] defines a risk functional in terms of the squared moment deviation and is thus related to GMM.

The role and interpretation of the "adversary" function $g \in \mathcal{G}$ and its associated objective function vary subtly across different saddle-point formulations, which in turn influences the types of assumptions required. This distinction is important: if $g$ is intended to approximate $\mathbb{E}\left[Y - h(X) \mid Z\right]$, then the space $\mathcal{G}$ must be sufficiently rich to do so, which is reflected via a closedness condition. If $g$ is primarily a Lagrange multiplier, its existence and properties are more directly tied to conditions like a source condition. This difference helps explain why [2] can dispense with the closedness assumption while other methods may require it.

**About the contrastive loss.** The contrastive loss used to learn spectral features derives its name from the foundational concept of contrastive learning, which traces back to the early 1990s. Notably, [5] introduced the Siamese network architecture for signature verification, which utilized a form similar to Eq.(8) to measure the distance between input pairs. Later, [10] formalized the contrastive loss function for face verification tasks. This loss is, in turn, linked to a truncated conditional operator (see Eq. (6)). The underlying principle has been frequently redeveloped under various names, such as correspondence analysis [17], principal inertia components [21, 22] for finite alphabets, the contrastive kernel [19, 12], and pointwise dependence [37] in self-supervised representation learning. The objective in Eq. (8) was also proposed and studied by [38] and used as a local approximation to the log-loss of classification deep neural networks [42]. More recently, [25] linked this same objective to the SVD of the conditional expectation operator.

# E   Experimental Details

In this section, we provide additional details for the experiments presented in the main text. Let $(Z, X)$ be generated as described in Section 6 with $n = 100000$. Figure 4 displays both the true data-generating density and the density corresponding to a set of learned spectral features. Even for $d = 11$, the resulting densities are complex. While it is feasible to conduct analogous experiments with higher values of $d$, we observed no qualitative changes in the outcomes. However, training and hyperparameter tuning became increasingly challenging as $d$ grew.

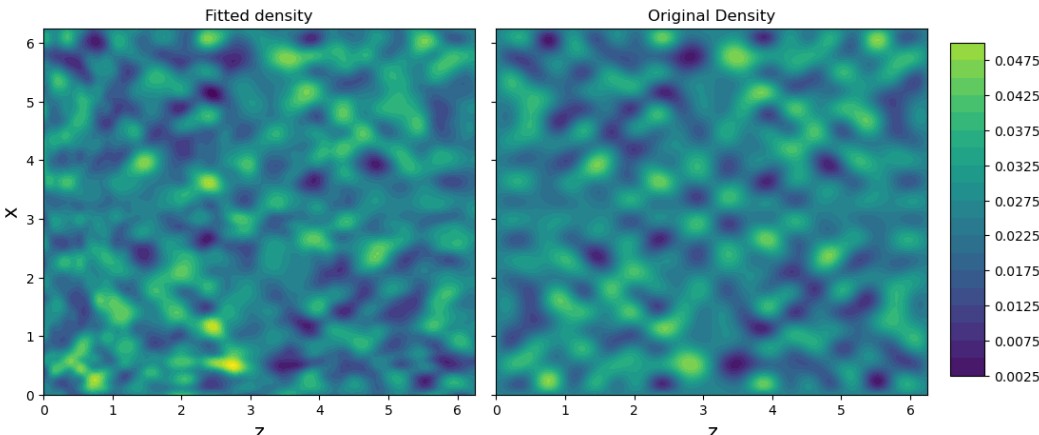

Figure 4: Comparison of a density corresponding to a set of learned spectral features **(left)** and the true data generating density **(right)**.

## E.1   Models Employed

The features were learned using two-hidden-layer neural networks. All models shared the same architecture, with layer widths $[1, 50, 50, 50]$: the input is one-dimensional, and the final layer outputs 50 learned features. To encourage the models to learn more oscillatory functions, the first layer used the activation $x \mapsto x + \sin^2(x)$, as introduced by [44], followed by GELU-activated hidden layers and a final linear layer.

We note that the first singular eigenfunctions of the conditional expectation operator $\mathcal{T}$ are always the constant functions (see Section 2). Therefore, we hard-code the constant feature $\mathbb{1}_{\mathcal{Z}} \otimes \mathbb{1}_{\mathcal{X}}$ into the model and restrict the following learned features to be mean-zero.

In addition, we included a regularization term to penalize both feature collinearity and large feature norms. This regularizer is a sample-based approximation of the quantity defined in Equation (10) of [25]:

$$\mathbb{E}\left[\|\varphi(X)\varphi(X)^{\mathsf{T}} - I\|^2\right] + \mathbb{E}\left[\|\psi(Z)\psi(Z)^{\mathsf{T}} - I\|^2\right] + 2\mathbb{E}\left[\|\varphi(X)\|^2\right] + 2\mathbb{E}\left[\|\psi(Z)\|^2\right].$$

The spectral features were trained on 100,000 samples of $(Z, X)$, while the 2SLS estimator built from the learned features used a separate dataset of 10,000 samples of $(Z, X, Y)$.

## E.2 Expanded example of the ugly scenario

As noted in the main text, the difficulty of recovering $h_0$ increases when it is not well supported on the singular functions of the conditional expectation operator $\mathcal{T}$. If the projection of $h_0$ onto the span of the learned singular functions is sufficiently large, the missing components may not significantly harm the quality of the estimate. We illustrate this property with a controlled experiment designed to vary the amount of support $h_0$ has in the singular space of $\mathcal{T}$.

We fix $d - 1$ orthonormal features $u_i$ for $Z$ and $v_i$ for $X$, and define the operator:

$$\mathcal{T} = \mathbb{1}_{\mathcal{Z}} \otimes \mathbb{1}_{\mathcal{X}} + \sum_{i=1}^{d-1} \sigma_i u_i \otimes v_i,$$

mirroring the setup from Section 6. We vary the spectrum by setting $\sigma_{1:k} = c$ and $\sigma_{k+1:d-1} = 0$ for $k \in \{1, \ldots, d-1\}$. We then define the target function:

$$h_0 = \frac{1}{\sqrt{d-1}} \sum_{i=1}^{d-1} v_i,$$

which is uniformly spread across the feature directions. As $k$ increases, the projection of $h_0$ onto the singular space of $\mathcal{T}$ increases in discrete steps from 0 to 1. This results in a corresponding qualitative improvement in the accuracy of the 2SLS estimator for $h_0$. Figure 5 illustrates this behavior: for small values of $k$, where $h_0$ is largely orthogonal to the singular functions of $\mathcal{T}$, the estimate fails to recover $h_0$. As $k$ grows, and more of $h_0$'s energy lies in the span of these singular functions, the 2SLS estimate increasingly aligns with the true function.

## F   Estimating spectral alignment

In this section, we present the details of the methodology introduced in Section 5. We recall the central formula provided in Eq. (10).

$$\langle v_i, h_0 \rangle_{L_2(X)} = \frac{1}{\sigma_i} \mathbb{E}[Y \cdot u_i(Z)]. \tag{16}$$

To evaluate spectral alignment, we shall approximate the RHS of the above. Suppose we have learned an operator $\widehat{\mathcal{T}}_d = \sum_{i=1}^{d} \hat{\psi}_i \otimes \hat{\phi}_i = \Psi \Phi^*$ which approximates the true conditional mean operator $\mathcal{T}$. Here $\Psi \colon \mathbb{R}^d \to L^2(Z)$ and $\Phi \colon \mathbb{R}^d \to L^2(X)$ are the maps sending $\alpha \overset{\Psi}{\mapsto} \sum_{i=1}^{d} \alpha_i \hat{\psi}_i$ and $\alpha \overset{\Phi}{\mapsto} \sum_{i=1}^{d} \alpha_i \hat{\phi}_i$. One can compute the SVD of $\widehat{\mathcal{T}}_d$ in two steps. We start with the derivations assuming access to population quantities. Let $C_Z$, and $C_X$ be the $d \times d$ population covariance matrices of the learned $Z$, and $X$ features, respectively. For any $z$ and $x$, we then introduce the whitened feature vectors $\psi_i'(z) = (C_Z^{-1/2} \hat{\psi}(z))_i$ and $\phi_i'(x) = (C_X^{-1/2} \hat{\phi}(x))_i$. The corresponding operators, $\Psi' = \Psi C_Z^{-1/2}$ and $\Phi' = \Phi C_X^{-1/2}$, are isometries. We can then write

$$\widehat{\mathcal{T}}_d = \Psi' C_Z^{1/2} C_X^{1/2} (\Phi')^*.$$

$C_Z^{1/2} C_X^{1/2}$ is a $d \times d$ matrix, let its SVD be $C_Z^{1/2} C_X^{1/2} = O \Sigma_{\phi,\psi} P^*$, where $\Sigma_{\phi,\psi}$ is a positive-definite diagonal matrix with diagonal entries $\sigma_{\phi,\psi,i}$. Now we can write

$$\widehat{\mathcal{T}}_d = (\Psi' O) \Sigma_{\phi,\psi} (\Phi' P)^*.$$

Finally letting $u_{\phi,\psi,i} = \Psi' O e_i$, and $v_{\phi,\psi,i} = \Phi' P e_i$, with the corresponding operators $U_{\phi,\psi} \colon \mathbb{R}^d \to L^2(Z)$, and $V_{\phi,\psi} \colon \mathbb{R}^d \to L^2(X)$, we get the SVD

$$\widehat{\mathcal{T}}_d = U_{\phi,\psi} \Sigma_{\phi,\psi} V_{\phi,\psi}^*.$$

In practice, given a $(Z, X)$ dataset, possibly the same one as was used to train $\widehat{\mathcal{T}}_d$, one can perform sample-based counterparts to the above procedures to get an approximate SVD.

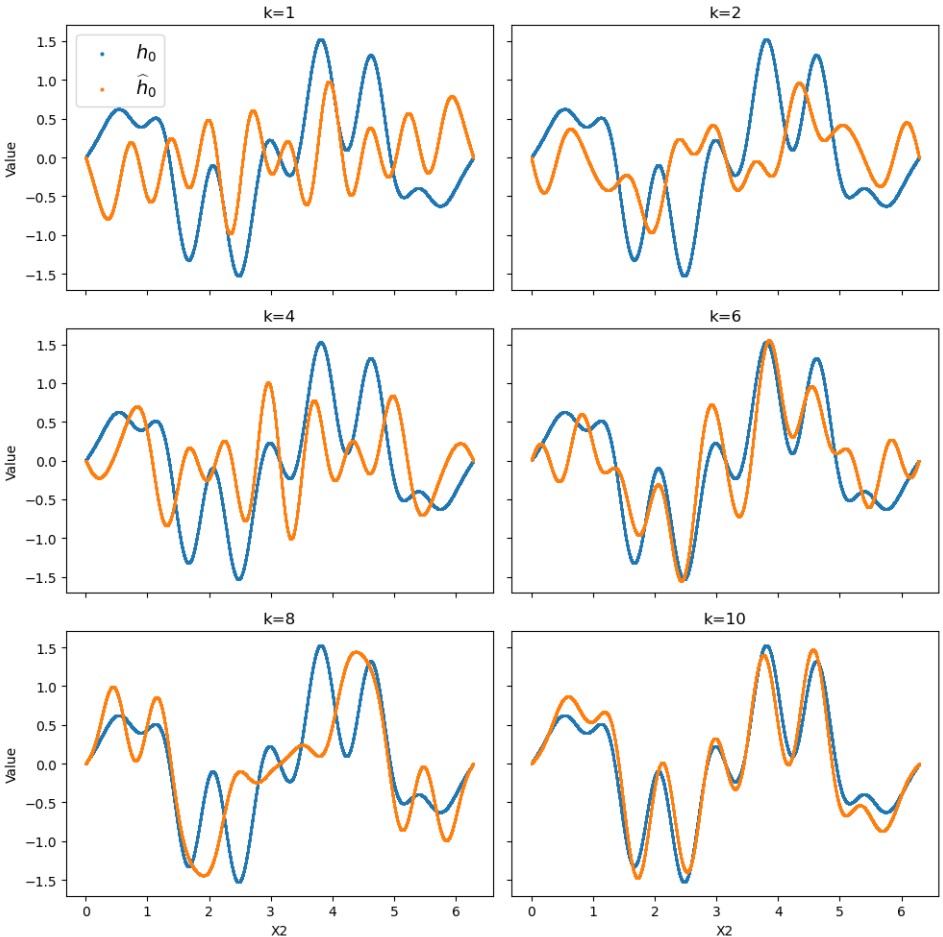

Figure 5: Qualitative improvement in the 2SLS estimate of $h_0$ as $k$ increases. When $k = 1$, $h_0$ is orthogonal to the singular functions of $\mathcal{T}$; when $k = d - 1$, it is fully contained in their span.

**Rationale for the alignment approximation method.** We are interested in evaluating the length of the projection of $h_0$ onto the leading learned singular functions of $\widehat{\mathcal{T}}_d$, that is of $\langle v_{\phi,\psi,i}, h_0 \rangle$. Following Eq. (10), we have

$$\langle v_{\phi,\psi,i}, h_0 \rangle = \sigma_{\phi,\psi,i}^{-1} \langle \widehat{\mathcal{T}}_d^* u_{\phi,\psi,i}, h_0 \rangle = \sigma_{\phi,\psi,i}^{-1} \langle u_{\phi,\psi,i}, \widehat{\mathcal{T}}_d h_0 \rangle$$
$$= \sigma_{\phi,\psi,i}^{-1} \langle u_{\phi,\psi,i}, \mathcal{T} h_0 \rangle + \sigma_{\phi,\psi,i}^{-1} \langle u_{\phi,\psi,i}, (\widehat{\mathcal{T}}_d - \mathcal{T}) h_0 \rangle.$$

The first term on the RHS above can be equivalently written as $\sigma_{\phi,\psi,i}^{-1} \langle u_{\phi,\psi,i}, \mathcal{T} h_0 \rangle = \sigma_{\phi,\psi,i}^{-1} \mathbb{E}[u_{\phi,\psi,i}(Z)Y]$, for which it is easy to compute a sample-based approximation. This approximation should be close to the true value of $\langle v_{\phi,\psi,i}, h_0 \rangle$ provided that the second RHS term is small. Letting $\mathcal{T}_d$ be the rank-$d$ SVD truncation of $\mathcal{T}$ we further telescope the second term on the RHS to get

$$\sigma_{\phi,\psi,i}^{-1} \left( \langle u_{\phi,\psi,i}, (\widehat{\mathcal{T}}_d - \mathcal{T}_d) h_0 \rangle + \langle u_{\phi,\psi,i}, (\mathcal{T}_d - \mathcal{T}) h_0 \rangle \right). \tag{17}$$

Since $\widehat{\mathcal{T}}_d$ should converge to $\mathcal{T}_d$ for a sufficiently flexible feature-learning model class, it is sensible to bound the norm of the first term above with $\sigma_{\phi,\psi,i}^{-1} \|\widehat{\mathcal{T}}_d - \mathcal{T}_d\| \|h_0\|$. For any $i \leq d$, this upper bound should converge to 0 with the size of the feature-learning sample size. To upper-bound the second term of Eq. (17), note that by Wedin Sin-$\Theta$ Theorem, we have

$$\|\Pi_{u_{1:d,\phi,\psi}} - \Pi_{u_{1:d}}\| \leq \frac{\|\mathcal{T}_d - \widehat{\mathcal{T}}_d\|}{\sigma_d}.$$

Thus

$$|\langle u_{\phi,\psi,i}, (\mathcal{T}_d - \mathcal{T})h_0\rangle| \leq \|\Pi_{u_{\phi,\psi,1:d}}(\mathcal{T}_d - \mathcal{T})h_0\| \leq \underbrace{\|\Pi_{u_{1:d}}(\mathcal{T}_d - \mathcal{T})h_0\|}_{=0} + \frac{\|\mathcal{T}_d - \widehat{\mathcal{T}}_d\|}{\sigma_d} \cdot \|(\mathcal{T}_d - \mathcal{T})h_0\|.$$

By construction, $\mathrm{Span}(u_{1:d})$ is orthogonal to the image of $\mathcal{T}_d - \mathcal{T}$. Hence the first term on the RHS vanishes, leaving us with

$$\left|\langle v_{\phi,\psi,i}, h_0\rangle - \sigma_{\phi,\psi,i}^{-1}\mathbb{E}[Yu_{\phi,\psi,i}]\right| \leq \sigma_{\phi,\psi,i}^{-1}\|\widehat{\mathcal{T}}_d - \mathcal{T}_d\|\|h_0\| + \sigma_d^{-1}\sigma_{\phi,\psi,i}^{-1}\|\mathcal{T}_d - \widehat{\mathcal{T}}_d\|\|(\mathcal{T}_d - \mathcal{T})h_0\|. \tag{18}$$

The essential takeaway from the bound above is that as long as the feature-learning sample size can increase to ensure $\|\widehat{\mathcal{T}}_d - \mathcal{T}_d\|$ is small, approximating the alignment with $\sigma_{\phi,\psi,i}^{-1}\mathbb{E}[Yu_{\phi,\psi,i}]$ is consistent. Moreover, the bound becomes tighter when,

1. $\sigma_{\phi,\psi,i}$ is big (close to 1), which corresponds to a slow spectral decay or looking at the top singular function.
2. $h_0$ lies in the top of the spectrum of $\mathcal{T}$ so that $\|(\mathcal{T}_d - \mathcal{T})h_0\|$ is small.

We can additionally compare how close is $\sigma_{\phi,\psi,i}^{-1}\mathbb{E}[Yu_{\phi,\psi,i}]$ to the alignment with the true eigenfunction of $\mathcal{T}$: $\langle v_i, h_0\rangle$, so that we can evaluate if we are in the "good" or "bad" scenario. For $i \leq d$, denote $\alpha_i = \langle v_{\phi,\psi,i}, h_0\rangle$ and $\hat{\alpha}_i = \sigma_{\phi,\psi,i}^{-1}\mathbb{E}[Yu_{\phi,\psi,i}(Z)]$. The previous bound shows that

$$|\alpha_i - \hat{\alpha}_i| \leq \mathrm{Err}_i, \qquad \mathrm{Err}_i = \sigma_{\phi,\psi,i}^{-1}\|\widehat{\mathcal{T}}_d - \mathcal{T}_d\|\|h_0\| + \sigma_d^{-1}\sigma_{\phi,\psi,i}^{-1}\|\mathcal{T}_d - \widehat{\mathcal{T}}_d\|\|(\mathcal{T}_d - \mathcal{T})h_0\|.$$

Then

$$\left|\|\Pi_{v_{\phi,\psi,1:d}}h_0\|^2 - \sum_{i=1}^d \sigma_{\phi,\psi,i}^{-2}|\mathbb{E}[Yu_{\phi,\psi,i}(Z)]|^2\right| = \left|\sum_{i=1}^d(\alpha_i^2 - \hat{\alpha}_i^2)\right| = \left|\sum_{i=1}^d(\alpha_i - \hat{\alpha}_i)(\alpha_i + \hat{\alpha}_i)\right|$$

$$\leq \sum_{i=1}^d |\alpha_i - \hat{\alpha}_i| \cdot |\alpha_i + \hat{\alpha}_i|$$

Next,

$$|\alpha_i + \hat{\alpha}_i| = |2\alpha_i - (\alpha_i - \hat{\alpha}_i)| \leq 2|\alpha_i| + |\alpha_i - \hat{\alpha}_i| \leq 2\|h_0\| + \mathrm{Err}_i,$$

and

$$\left|\|\Pi_{v_{\phi,\psi,1:d}}h_0\|^2 - \sum_{i=1}^d \sigma_{\phi,\psi,i}^{-2}|\mathbb{E}[Yu_{\phi,\psi,i}(Z)]|^2\right| \leq \sum_{i=1}^d \mathrm{Err}_i \cdot (2\|h_0\| + \mathrm{Err}_i)$$

$$= 2\|h_0\|\sum_{i=1}^d \mathrm{Err}_i + \sum_{i=1}^d(\mathrm{Err}_i)^2$$

Finally, to assess how well we get the alignment to the eigenfunctions of $\mathcal{T}$ observe that

$$\left|\|\Pi_{v_{\phi,\psi,1:d}}h_0\| - \|\Pi_{v_{1:d}}h_0\|\right| \leq \left\|\left(\Pi_{v_{\phi,\psi,1:d}} - \Pi_{v_{1:d}}\right)h_0\right\| \leq \frac{\|\mathcal{T}_d - \widehat{\mathcal{T}}_d\|}{\sigma_d}\|h_0\|.$$

**Practical considerations.** In practice, we want to perform sample-based approximations of the procedure described above, and while the top singular functions of $\mathcal{T}$ and the corresponding singular values can be reliably estimated, learning the bottom of the spectrum is more unreliable. Given that we divide $\mathbb{E}[Yu_{\phi,\psi,i}(Z)]$ by $\sigma_{i,\phi,\psi}$, for singular values close to 0, a small error in estimating them is inflated by the inversion. Hence, we resort to a heuristic that allows us to decide which features and singular functions are learned reliably. Let $\widehat{\mathcal{T}}_d = \widehat{U}_{\phi,\psi}\widehat{\Sigma}_{\phi,\psi}\widehat{V}_{\phi,\psi}$ be a finite-sample approximation of the SVD of $\widehat{\mathcal{T}}_d^*$. That is, we perform the SVD computation procedures of the preceding paragraph but relying on sample feature covariance matrices $\widehat{C}_X$ and $\widehat{C}_Z$. After fitting the SVD once, we recompute the covariance of $\widehat{u}_{\phi,\psi,i}$ and $\widehat{v}_{\phi,\psi,i}$ on resampled $(Z^{(b)}, X^{(b)})$ data with $b = 1, ..., B$ and extract its diagonal terms, which we refer to as $\hat{\sigma}_{\phi,\psi,1:d}^{(b)}$. Letting the original singular value estimates be

$\hat{\sigma}_{\phi,\psi,i} \doteq \hat{\sigma}^{(0)}_{\phi,\psi,i}$, we can use the sets $(\hat{\sigma}^{(b)}_{\phi,\psi,i})^B_{b=0}$ to evaluate the reliability of a given singular value estimate. If the variance of the estimates in a given collection of $\hat{\sigma}^{(b)}_{\phi,\psi,k}$ is large relative to their mean, we treat the value as unreliable and discard all the singular values $\sigma_{\phi,\psi,i \geq k}$ from further analysis.

For the remaining singular values, $\hat{\sigma}_{\phi,\psi,1}, \cdots, \hat{\sigma}_{\phi,\psi,k}$, which we treat as "reliably learned" we utilise the quantiles of bootstrap samples to construct pseudo-confidence-sets $[\hat{\sigma}^{\min}_{\phi,\psi,i}, \hat{\sigma}^{\max}_{\phi,\psi,i}]$ for $\sigma_{\phi,\psi,i}$. We use these as proxies for upper and lower bounds on $\sigma_{\phi,\psi,i}$. These provide us with matching pseudo-bounds on our estimates of $\langle v_{\phi,\psi,i}, h_0 \rangle$. That is, the "confidence set" for $\langle v_i, h_0 \rangle$ is the interval

$$\left[ (\hat{\sigma}^{\max}_{\phi,\psi,i})^{-1} \widehat{\mathbb{E}}[Y \hat{u}_{\phi,\psi,i}(Z)], (\hat{\sigma}^{\min}_{\phi,\psi,i})^{-1} \widehat{\mathbb{E}}[Y \hat{u}_{\phi,\psi,i}(Z)] \right],$$

together with the central estimate $(\hat{\sigma}^{(0)}_{\phi,\psi,i})^{-1} \widehat{\mathbb{E}}[Y \hat{u}_{\phi,\psi,i}(Z)]$.

### F.1 Spectral alignment in synthetic examples

Note that whenever the structural function is available, as is the case in our one-dimensional synthetic example and in the dSprites experiment, one can evaluate the correctness of the approximation proposed above by comparing it to $\langle h_0, v_{\phi,\psi,i} \rangle$. In Figure 6, we observe that the method is reliable if the decay of singular values is sufficiently slow for feature-learning to pick up on the singular function pairs reliably. For $c_\sigma = 0.8$ the approximations are very good while for $c_\sigma = 0.2$ they become entirely unreliable. A more thorough theoretical analysis of these estimates is a topic warranting further investigation but these observations are generally in line with Eq. (18). In particular, small values of $\sigma_{\phi,\psi,i}$ inflate the approximation error.

### F.2 dSprites is in the good regime

We fitted the spectral-feature models for dSprites using the same architectures as proposed by [36]. Analogously to the fully synthetic experiment, we are able to evaluate the spectrum of $\widehat{\mathcal{T}}$ and directly measure where in it, $h_0$ is supported. As seen in Figure 7, the top of the learned spectrum is very flat, and hence, in line with what we observed in the previous example, the alignment estimates are reliable. Our experiments confirm that $h_0$ is usually spanned by the leading 32 spectral features and 85% of its squared norm spanned by the leading 10 functions. Hence, dSprites is indeed an example of the good case.

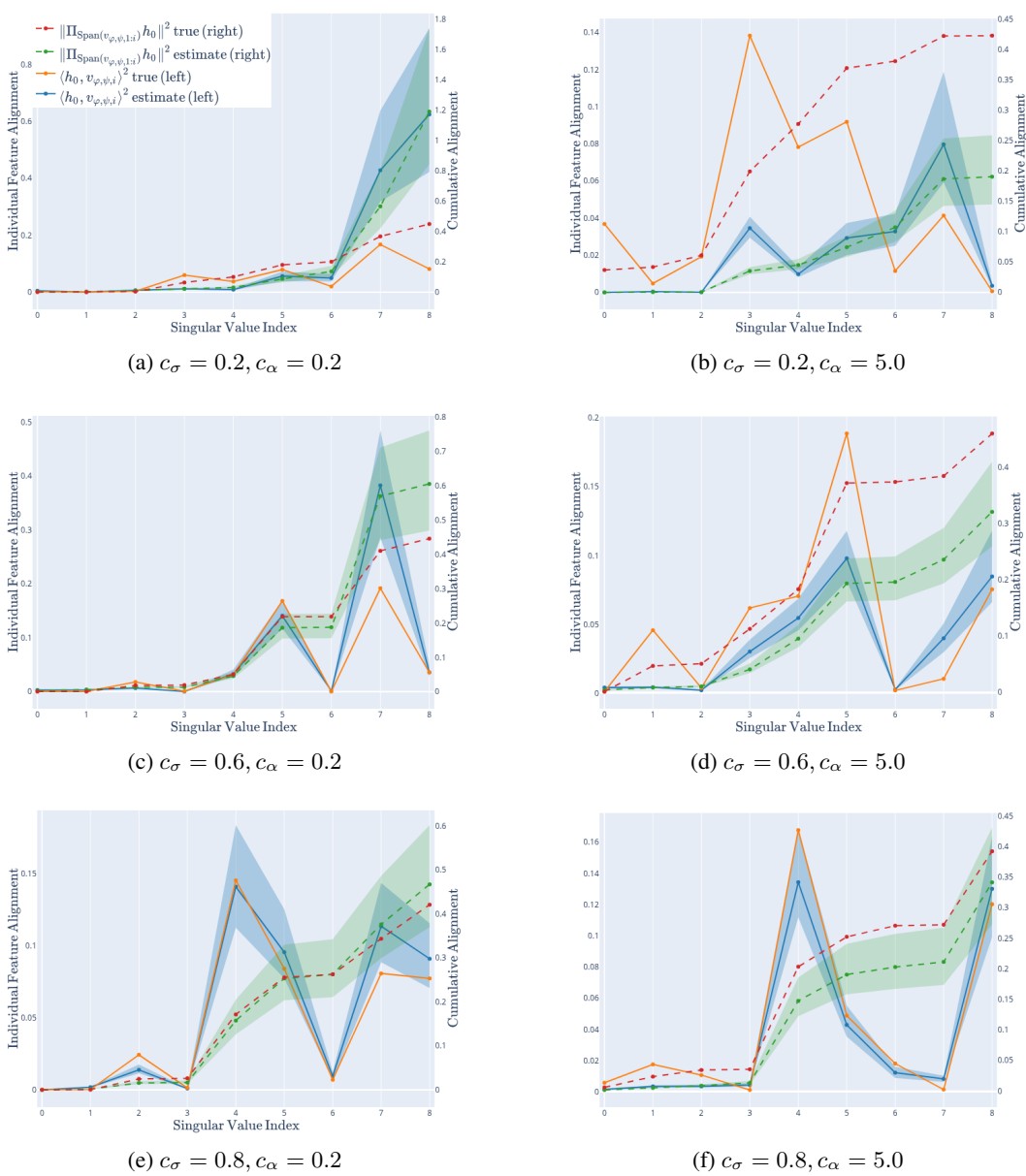

Figure 6: Evaluation of the reliability of spectral alignment estimation depending on the real spectral alignment measured in terms of $c_\alpha$ and the rate of singular value decay $c_\sigma$.

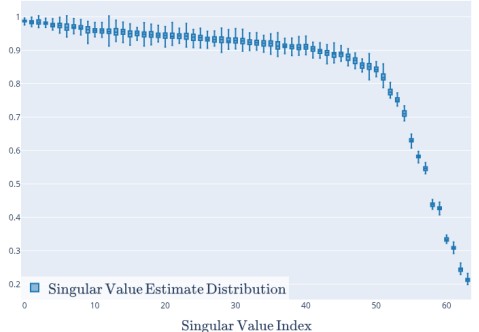
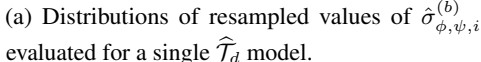

(a) Distributions of resampled values of $\hat{\sigma}_{\phi,\psi,i}^{(b)}$ evaluated for a single $\widehat{\mathcal{T}}_d$ model.

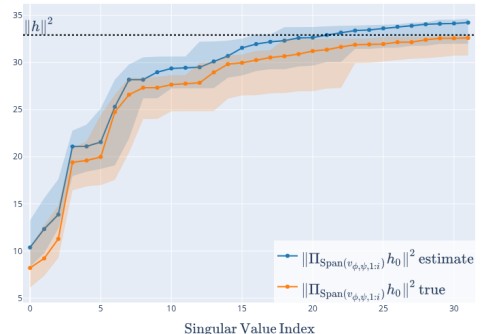

(b) Estimates and true values of the projection length of $h_0$ onto the leading $i$ features.

Figure 7: Evaluation of dSprites spectral alignment. Left: the confidence intervals correspond to bootstrapped refits of the feature covariance for an individual model. Right: the confidence intervals are obtained from 9 independently trained models with identical hyperparameters but different random seeds.

