# OpenReview forum: "Demystifying Spectral Feature Learning for Instrumental Variable Regression"
_NeurIPS.cc/2025/Conference — NeurIPS 2025 poster_

### Official Review · Reviewer_QU6R · 2025-06-12

**Clarity:** 3
**Significance:** 3
**Originality:** 2
**Rating:** 4
**Confidence:** 3

**Summary:**

This paper learns spectral features for NPIV regression to estimate causal effects in the presence of hidden confounders. It derives a generalization error bound for a 2SLS estimator utilizing these features, identifying three performance regimes: good, bad, and ugly.

**Questions:**

1. Provide a dedicated subsection (e.g., in the related work or methodology) that offers a detailed comparative analysis with "Kernel Instrumental Variable Regression" (Singh et al., 2019), "Dual Instrumental Variable Regression" (Bennett et al., 2023), and "Fast Instrument Learning with Faster Rates" (Kim et al., 2022). While some of these works are cited, the current discussion is too brief and appears to misrepresent aspects of their original formulations.

2. Could the authors expand the experimental validation to include comparisons with a broader range of advanced instrumental variable methods (e.g., Kernel IV, Dual IV, AGMM, DeepGMM, Fast IV, DFIV, DeepIV) and, ideally, incorporate experiments on at least one high-dimensional, complex real-world dataset?

**Ethical Concerns:**

["NO or VERY MINOR ethics concerns only"]

**Final Justification:**

Authors agreed to expand the theoretical comparison and add dSprites experiments, partially addressing my concerns. However, the lack of a direct head-to-head comparison with Kernel IV, Dual IV, AGMM, DeepGMM, Fast IV, DFIV, and DeepIV persists. I therefore raise my score but not to be 5/6.

**Quality:**

2

**Strengths And Weaknesses:**

**Strengths:**

1. The paper studies nonparametric instrumental variable regression, a critical problem in causal effect estimation where hidden confounders exist.

2. The paper derives a generalization error bound for a two-stage least squares estimator based on spectral features.

3. While spectral features have been heuristically linked to NPIV in prior work , this paper provides a novel and rigorous theoretical characterization of their performance through the generalization error bound and the "good-bad-ugly" taxonomy.

**Weakness:**

1. The paper acknowledges that instrumental variable regression is often implemented as a two-stage procedure and mentions fixed and learned features. However, while it references other works that formulate IV regression as saddle-point optimization , it could provide a more detailed comparative analysis of its proposed method with specific, related prior works such as "Kernel Instrumental Variable Regression" (Singh et al., 2019 ), "Dual Instrumental Variable Regression" (Bennett et al., 2023 ), and "Fast Instrument Learning with Faster Rates" (Kim et al., 2022 ). Explicitly highlighting the similarities, key differences, and unique advantages or limitations compared to these methods would strengthen the paper's positioning.

2. The experimental section is simple, primarily relying on synthetic data where feature spaces are modeled using trigonometric functions. Its generalizability to highly complex, real-world, high-dimensional data (e.g., images, text) where neural networks are typically employed for feature learning is not fully explored or demonstrated.

3. The experimental section would be significantly strengthened by including comparisons with state-of-the-art methods such as Kernel IV, Dual IV, AGMM, DeepGMM, Fast IV, DFIV, and DeepIV.

4. The synthetic experiments use trigonometric functions for feature spaces. While useful for isolating effects, the generalizability of these findings to highly complex, high-dimensional real-world data where neural networks are typically employed might warrant further discussion or empirical demonstration.

5. The main body of the paper lacks a dedicated "Conclusion" section. Additionally, while the authors state in their checklist that they discuss the limitations of their work, this discussion is not apparent in the corresponding sections of the paper.

---

> ### Author Rebuttal · Authors · 2025-07-30
>
> We thank the reviewer for their careful reading and insightful comments on our paper. We have addressed the reviewer concerns and questions below, and we believe our proposed revisions strengthen the manuscript.
>
> We would be grateful if the reviewer might consider increasing their score in case they are satisfied with the proposed changes.
>
> - **Weakness 1 & Question 1. Detailed comparative analysis in literature review**. We agree that given the large amount of NPIV papers, a more explicit discussion of the similarities, key differences, and unique theoretical advantages or limitations of our approach compared to these relevant methods would strengthen the paper's positioning within the literature. We will revise the related work in the camera-ready version to include this expanded theoretical comparative analysis and the missing citation to "Dual Instrumental Variable Regression" and "Fast Instrument Learning with Faster Rates".
>
> - **Weakness 3 & Question 2. Experimental comparisons with state-of-the-art methods**. It is important to clarify that our paper primarily focuses on the theoretical characterization of spectral features and their performance regimes (good, bad, ugly) for NPIV. The spectral features framework, which our paper analyzes theoretically, has already been empirically benchmarked extensively against a range of other state-of-the-art methods by [1] (including KIV, DFIV, DeepGMM and Dual Embedding). That study provides comprehensive empirical evidence of its practical performance against established benchmarks. Therefore, a full-scale empirical comparison was not the main objective of the present theoretical contribution. We will add a sentence to the experimental section of the main paper to explicitly direct readers to this relevant  work and its comprehensive benchmarks.
>
> - **Weaknesses 2 & 4. Experimental generalizability to complex, high-dimensional data.** We agree that our experimental section could be expanded to better illustrate generalizability from synthetic data to complex, high-dimensional, real-world settings where neural networks are typically employed.
>
>     To address this, our paper now includes empirical results on the dSprites dataset. This is a well-established, high-dimensional dataset for NPIV, serving as a crucial example of a complex scenario where our method (which uses neural networks for feature learning) demonstrates strong applicability and scalability. We refer to our answer to W1 of **Reviewer m9NX** for further details. Furthermore, the spectral contrastive learning approach has previously achieved state-of-the-art results on challenging Demand Design datasets [1], further proving its efficacy on complex real-world data.
>
>     While our synthetic experiments use trigonometric functions to clearly isolate theoretical effects, we believe our new results on dSprites and the performance on Demand Design in prior work provide strong evidence for the generalizability of our findings to more complex settings.
>
> - **Weakness 5. Missing conclusion section.** We will use the extra page given for the camera ready version to add a concluding discussion and perspective on future works. We acknowledge that key limitations of our submitted work included the absence of a strategy to empirically quantify spectral alignment and decay for identifying the "good, bad, or ugly" scenarios, and the lack of a precise theoretical characterization of the error term $\varepsilon$ in Proposition 3. We have addressed the former through a newly developed empirical strategy for spectral alignment, as elaborated in our response to W1 for **Reviewer m9NX**. We also recognize the latter as an important and open problem for future research, which we briefly discussed on page 8 of our manuscript and further elaborate on in our response to W3 for **Reviewer m9NX**. We will explicitly discuss these limitations in the conclusion.
>
> [1] Haotian Sun, Antoine Moulin, Tongzheng Ren, Arthur Gretton, Bo Dai "Spectral Representation for Causal Estimation with Hidden Confounders", AISTATS, 2025.

---

> > ### Comment · Reviewer_QU6R · 2025-08-05
> >
> > Thank you for the clarification. I will raise my score on the promise that the authors will (1) extend the discussion of recent NPIV baselines and (2) add experiments on the dSprites dataset. One suggestion is to directly include these baselines—Kernel IV, Dual IV, AGMM, DeepGMM, Fast IV, DFIV, and DeepIV—as comparisons in the main paper

---

> > > ### Author Response · Authors · 2025-08-07
> > >
> > > Thank you for your feedback and for taking the time to review our paper. We appreciate your engagement and are glad the clarifications were helpful. We confirm that we will address points (1) and (2) in the camera-ready version.

---

### Official Review · Reviewer_baed · 2025-06-29

**Clarity:** 2
**Significance:** 3
**Originality:** 3
**Rating:** 4
**Confidence:** 2

**Summary:**

This paper studies the problem of causal effect estimation under hidden confounding using nonparametric instrumental variable regression. The authors propose a two-stage least squares estimator based on spectral features derived from the conditional expectation operator. A key contribution is the theoretical analysis that establishes a generalization error bound and highlights two critical factors affecting performance: the spectral alignment between the structural function and the operator’s eigenfunctions, and the decay rate of the operator’s singular values, which reflects instrument strength. Based on these insights, the paper introduces a good–bad–ugly taxonomy that characterizes when the method performs well or fails. Synthetic experiments are carefully designed to validate this taxonomy.

**Questions:**

Add more comparison experiments: One notable limitation of the experimental section is the absence of comparisons with established baseline methods such as Kernel IV or DeepIV. These kernel-based and deep learning-based IV models are commonly used in causal inference under endogeneity and provide strong empirical baselines. Adding such comparisons would make the evaluation more convincing by showing where spectral feature learning performs better or worse in more realistic and challenging scenarios. Without these baselines, it is difficult to tell whether the proposed method brings practical improvements over existing approaches, especially when the spectral alignment or properties of the operator cannot be directly observed or controlled.

**Ethical Concerns:**

["NO or VERY MINOR ethics concerns only"]

**Limitations:**

Yes.

**Quality:**

3

**Strengths And Weaknesses:**

S1: The paper provides a rigorous theoretical analysis of spectral 2SLS for NPIV regression, including provable generalization bounds.

S2: The experiments are carefully designed, I like the setting for good–bad–ugly taxonomy.

S3: The integration of spectral learning theory with causal IV estimation is novel

W1: While the theoretical results are solid, no real-world datasets are used to validate the findings.

W2: The paper occasionally assumes familiarity with advanced spectral operator theory, which may be difficult for broader NeurIPS audiences (from my point of view).

W3: The  practical applicability is somewhat limited by the fact that real data rarely comes with ground truth for spectral alignment or singular value decay, making it hard to operationalize the taxonomy without strong assumptions.

---

> ### Author Rebuttal · Authors · 2025-07-30
>
> We thank the reviewer for their careful reading and positive feedback on our paper. We have addressed the reviewer concerns and questions below, and we believe our proposed revisions significantly strengthen the manuscript.
>
> We would be grateful if the reviewer might consider increasing their score in case they are satisfied with the proposed changes.
>
> - **W1. Lack of real-world datasets.** We agree with the reviewer that incorporating more complex datasets would strengthen our experimental validation. To address this, we have added empirical results on the dSprites dataset. This is a well-established, high-dimensional dataset for NPIV, serving as a crucial example of a complex scenario where our method demonstrates strong applicability and scalability. We refer to our answer to Weakness 1 of **Reviewer m9NX** for details.
>
> - **W2. Assumption of familiarity with advanced spectral operator theory.** We agree that parts of the paper might assume a high degree of familiarity with advanced spectral operator theory, potentially limiting accessibility for a broader NeurIPS audience. To address this, we will add a dedicated section in the appendix to provide a concise and accessible introduction to the necessary concepts from spectral operator theory. This section will clarify key definitions and notations essential for understanding our theoretical framework, aiming to make the paper more self-contained and interpretable without burdening the main text.
>
> - **W3. Practical applicability and unobservable spectral properties.** We agree that the unobservability of these properties in practice is a significant challenge. Crucially, since submission, we have developed a strategy to empirically quantify spectral alignment and eigendecay and have tested it on the dSprites dataset. We also show how to empirically identify the effective cutoff dimension ($d$). This provides a means to gauge the decay of singular values and thus the effective ill-posedness, helping to operationalize aspects of the "good," "bad," and "ugly" scenarios in practice. We refer to our answer to Question 2 & Weakness 1 of **Reviewer m9NX** for details.
>
>     We will incorporate details of these empirical strategies into the paper, demonstrating how practitioners can gain insight into the spectral properties of their data, even without ground truth.
>
> - **Question. Absence of comparisons with established baseline methods.** It is important to clarify that our paper primarily focuses on the theoretical characterization of spectral features and their performance regimes (good, bad, ugly) for NPIV. The spectral features framework, which our paper analyzes theoretically, has already been empirically benchmarked extensively against a range of other state-of-the-art methods by [1] (including KIV, DFIV, DeepGMM and Dual Embedding). That study provides comprehensive empirical evidence of its practical performance against established benchmarks. Therefore, a full-scale empirical comparison was not the main objective of the present theoretical contribution. We will add a sentence to the experimental section of the main paper to explicitly direct readers to this relevant work and its comprehensive benchmarks.
>
> [1] Haotian Sun, Antoine Moulin, Tongzheng Ren, Arthur Gretton, Bo Dai "Spectral Representation for Causal Estimation with Hidden Confounders", AISTATS, 2025.

---

> > ### Comment · Reviewer_baed · 2025-08-05
> >
> > I would like to thank the authors for their detailed and thoughtful response. However, due to my low confidence in spectral theory, I have decided to keep my original score.

---

> > > ### Author Response · Authors · 2025-08-07
> > >
> > > Thank you for your feedback and for taking the time to engage with our work. We appreciate your honesty regarding your confidence in spectral theory. We hope that the additional background and clarifications on spectral methods, which we plan to include in the appendix, will help make our contributions more accessible and appreciated by a broader audience.

---

### Official Review · Reviewer_m9NX · 2025-07-01

**Clarity:** 4
**Significance:** 4
**Originality:** 3
**Rating:** 5
**Confidence:** 3

**Summary:**

The paper provides a theoretical analysis of the spectral feature learning for instrumental variable regression. Via a detailed analysis of the conditional expectation operator $\mathcal{T}$ between the features and the instrument, the work complements previous empirical observations that the performance of two stage IV regression estimator depends on the spectral structure of the operator and its alignment with the ground-truth link function. The paper shows that spectral contrastive learning, introduced in previous work, approximates the SVD of $\mathcal{T}$ and identifies three regimes which determine the effectiveness of such spectral-based methods. These regimes depend on the spectral alignment and singular value decay of the conditional expectation operator. Finally, the authors illustrate their theoretical results and the identified regimes with numerical simulations, where they explicitly control the decay and alignment parameters.

**Questions:**

1. How robust are the results to "slight" violations of Assumption 1? Partial identifiability frequently arises in applications where one uses instrumental variable methods, cf. , for instance, anchor regression [1]. What would the error decomposition roughly look like in such a scenario?
2. In Proposition 3, the choice of the cutoff dimension $d$ seems to depend on the decay of singular values of $\mathcal{T}$ which is a priori unknown. How would one choose $d$ empirically?

[1] Rothenhäusler, Dominik, Nicolai Meinshausen, Peter Bühlmann, and Jonas Peters. "Anchor regression: Heterogeneous data meet causality." Journal of the Royal Statistical Society Series B: Statistical Methodology 83, no. 2 (2021): 215-246.

**Ethical Concerns:**

["NO or VERY MINOR ethics concerns only"]

**Final Justification:**

The authors' rebuttal has clarified several aspects of the paper and improved my understanding of its contributions. I believe that the added discussion of the empirical evaluation of spectral alignment improves the scope of the paper. Thus, my concerns have been largely addressed, and I would recommend the paper for acceptance.

**Limitations:**

The authors have properly addressed the limitations of the study.

**Quality:**

3

**Strengths And Weaknesses:**

Strengths:

1. The paper presents novel theoretical contributions: it shows that the current state-of-the art NPIV estimator, based on a contrastive learning objective, approximates the spectral structure of the conditional expectation operator. Based on this, the authors propose a useful and intuitive decomposition of its generalization error and propose regimes based on its behaviour. Importantly, the results seem to suggest that the empirical success of the state-of-the art 2SLS estimator does not translate into universal theoretical guarantees, but rather only illustrates the "good" regime, which is a valuable insight for theorists and practitioners in the field.
2. The paper is very well-written, self-contained and is very clear about its contributions and limitations.
3. All assumptions and results are appropriately introduced and additionally discussed informally, providing intuition.

Weaknesses:

1. The main weakness of the paper seems to be the experimental evaluation. For instance, in lines 92-94, it is suggested that the empirical success of the decoupled spectral approach in [32] is due to the experiments in that work satisfying the "good" spectral scenario. However, this claim is not backed by evaluation of any "proxies" for alignment/decay on the datasets in [32] (dSprites and Demand Design) placing them in a "good" scenario. Additionally, the work would profit from real-data or semi-synthetic experiments illustrating the bad/ugly scenarios, since this would provide evidence that those scenarios do occur in realistic datasets.
2. If I am not missing anything, the work does not seem to provide corresponding lower bounds for the generalization error. This would be useful, for instance, to prove optimality of spectral methods in the "good" regime. Since the upper bound in, e.g., Theorem 1, is not provably tight, the reader cannot conclude whether the three scenarios provide a fundamental limitation for Sieve 2SLS, information-theoretic limitations for any method or are artifacts of the upper bound. Thus, a tight analysis would greatly improve the interpretability of the scenarios.
3. In Proposition 3, the $\epsilon$ is defined implicitly, and the decay rate of the resulting $\frac{\epsilon}{\sigma_d}$ is unclear. This slightly weakens the interpretability of the approximation error upper bound.

---

> ### Author Rebuttal · Authors · 2025-07-30
>
> We thank the reviewer for their positive assessment of our contribution and valuable comments! We have addressed all points of concern and have proposed edits to the manuscript. We elaborate on the details below.
>
> We would be grateful if the reviewer might consider increasing their score in the event that they are satisfied with the proposed changes.
>
> - **Q1. Robustness to violations of Assumption 1.** As noted, estimating the structural function ($h_0$) in the NPIV model is equivalent to solving the ill-posed inverse problem $\mathcal{T}(h) = \mathbb{E} [Y∣Z]$ in $h$, where identifiability of $h_0$ hinges on the injectivity of the operator $\mathcal{T}$ (Darolles et al [9]). When $\mathcal{T}$ is not strictly injective, we face partial identifiability. In such scenarios, while unique identification of $h_0$ is lost, it is possible to establish consistency to a minimum-norm solution of the inverse problem. This approach is common in partially identified instrumental variable models (Bennett et al [2], Meunier et al [27]]).
>
>     Regarding the second part of Assumption 1 (existence of a solution, $\mathcal{T}^{−1}(\{\mathbb{E}[Y∣Z]\}) \ne \emptyset$): This is a fundamental model specification assumption. A violation here implies that our NPIV model structure is misspecified for the data, meaning no true $h_0$ exists that satisfies the moment condition of the model. This is a much more severe form of "violation" than a lack of injectivity, as it implies the problem itself has no solution consistent with the model, and therefore cannot be "slightly" violated in a meaningful way for consistent estimation. Any prior work on NPIV relies on this assumption for the existence of a suitable $h_0$.
>
>     Finally, we thank the reviewer for pointing us to anchor regression. It is crucial to clarify that anchor regression addresses a different challenge than violations of our Assumption 1. While our assumption pertains to the existence and injectivity of the operator $\mathcal{T}$ for the identifiability and consistent estimation of the structural function $h_0$ in the NPIV model, anchor regression is designed for scenarios where traditional instrumental variable assumptions, specifically the exclusion restriction, are violated (i.e., the instrument directly causes the outcome). This violation does lead to a loss of identifiability for the causal effect in classical IV setups. However, anchor regression's primary objective is distributionally robust prediction, providing a stable predictive model even when an 'instrument' is technically invalid, rather than focusing on the consistent identification of the underlying structural function as our NPIV framework does.
>
>     For conciseness, we did not elaborate on this in the main text. However, we agree this is an important nuance and will add a remark after Assumption 1 to clarify this point and its implications for partial identifiability.
>
> - **Q2. Empirical choice of cutoff dimension (d).** We agree that the optimal choice of the latent dimension $d$ (cutoff dimension) inherently depends on the decay of the operator's singular values, which is *a priori* unknown. This is a common challenge in ill-posed inverse problems and spectral methods, analogous to selecting the effective rank in noisy matrix completion. Theoretically, this can be addressed using concepts like the Baik–Ben Arous–Péché (BBP) transition [R1], which provide a formal prediction for the "edge" of the spectrum that arises purely from statistical noise. Any singular value found in the data that is significantly larger than this theoretical edge would be considered part of the signal. A simple and practical way to find this same signal-noise boundary is to perform a stability check: Repeatedly estimate the singular values on different bootstrap samples of the data. The estimates of the true "signal" singular values will be stable (low variance), while the "noise" values will be unstable (high variance). Then, choose $d$ at the "elbow" where this transition occurs. We refer to our answer to W1 below for further discussion.
>
> - **W1. Experimental evaluation, spectral scenarios.** Empirically evaluating proxies for spectral alignment is indeed essential for assessing whether practical examples fall into the "good" spectral scenario.
>
>     **Quantifying spectral alignment in dSprites**
>
>     After submission, inspired by the numerical methods for linear inverse problems [R2], we developped a strategy to quantify the spectral alignment, $a_i = \langle h_0 ,u_i\rangle_{L^2(X)}$ where $u_i$ are the left eigenfunctions of $\mathcal{T}$: $\mathcal{T} = \sum_{i \geq 1} \sigma_i v_i \otimes u_i$. Coeficient $a_i$ directly measures the contribution of the $i$-th singular function to $h_0$ in the $L_2(X)$-norm, that is $\lVert h_0 \rVert_{L_2(X)} = \sum_i a_i^2$.  Since $h_0$ is unknown, we exploit the identity:
>     $$
>     a_i = \frac{1}{\sigma_i} \langle h_0, \mathcal{T}^*v_i \rangle_{L_2(X)} = \frac{1}{\sigma_i} \langle \mathcal{T} h_0, v_i \rangle_{L_2(Z)} = \frac{1}{\sigma_i} \mathbb{E}[(\mathcal{T}h_0)(Z)v_i(Z)] =  \frac{1}{\sigma_i} \mathbb{E}[\mathbb{E}[Y \mid Z]v_i(Z)] =  \frac{1}{\sigma_i} \mathbb{E}[Yv_i(Z)]
>     $$
>     where we used the singular function relationship $\sigma_i u_i = \mathcal{T}^*v_i$ and the fact that $h_0$ satisfies $\mathcal{T}h_0=\mathbb{E}[Y \mid Z]$.
>
>     This identity shows we can estimate the coefficient $a_i$ by estimating the singular value $\sigma_i$ and the left singular function $v_i$ from $(Z, X)$ samples, and then estimate $a_i$ using $(Y, Z)$ samples. More specifically, the procedure is as follows.
>     1. Learn spectral features using the spectral contrastive method from our paper.
>     2. Estimate SVD components: Perform a canonical correlation analysis (CCA) on these features to get estimates of the leading singular values $\hat \sigma_i$ and singular function $\hat v_i$.
>     3. Quantify stability: Perform subsampling and get a measure of uncertainty for our estimates (e.g., define confidence sets using quantiles). One would typically expect the estimates for the leading components to be stable, while uncertainty grows for components corresponding to smaller singular values.
>     4. Estimate alignment: using these stable estimates, calculate the proportion of the $L_2$ norm of $h_0$ captured by the leading features.
>
>     Applying this procedure to the dSprites dataset reveals:
>     - The singular value spectrum is initially flat and then decays sharply, indicating a spectral structure that exhibits the BBP-type transition discussed in our response to Q2.
>     - Our alignment estimates show that approximately 85% of the squared norm of $h_0$ is explained by the top 10 features, and about 95% by the top 20.
>
>     Together, these findings provide evidence that the dSprites experiment falls into the "good" regime. This methodology provides a practical tool for assessing spectral properties in real-world settings. We will add a full description of this procedure and the new results to the appendix.
>
>     **Demand Design dataset**
>
>     The Demand Design dataset includes observed confounding variables. This leads to the conditional mean operator being non-compact. Consequently, direct notions of spectral alignment and decay for $\mathcal{T}$ become significantly more involved and are outside the scope of our current work, though an interesting and important problem for future research.
>
>     **Illustrating bad and ugly scenarios**
>
>     Illustrating the "bad" and "ugly" scenarios on real datasets is an important direction. Our primary goal in this work was to first establish a clear theoretical framework for these failure modes and validate their existence in a controlled setting. The empirical tools we have now developped (as discussed for dSprites) make it feasible to search for these scenarios in practice.
>
>     While we have not yet run specific experiments, we can describe plausible bad/ugly real-world situations:
>     - "Bad" scenario (weak instrument): this is a well-known challenge in applied econometrics, the bad regime can arise when an instrument has only a weak effect on the treatment. For instance, using small, local policy variations as instruments for nationwide economic behavior often leads to this issue.
>     - "Ugly" scenario: this failure mode can occur if the instrument influences the treatment through a mechanism that is irrelevant to the structural function. For example, if an instrument (like an insurance plan) predicts treatment (a drug), but the health outcome depends on a factor (like a gene) that is uncorrelated with the instrument, then the estimation of the structural function would fail because the instrument provides the "wrong" kind of variation in the treatment.
>
> - **W2. Lower bound.** We agree with the point made on the lower bound. As explained in our answer to Q1 of **Reviewer 9cMm**, we can now show that our upper bound in Corollary 1 is tight, addressing the optimality of spectral methods and the fundamental nature of our proposed scenarios. This will be added as a remark, with full details in the appendix.
>
> - **W3. Control on $\varepsilon$.** We agree that the absence of precise characterization of $\varepsilon$ in Proposition 3 weakens interpretability. However, representation learning is challenging enough on its own, and uncovering tight theoretical bounds for $\varepsilon$ would require a dedicated and substantial investigation (e.g., a full paper on spectral contrastive learning generalization), which lies beyond the scope of our current work. We will add a remark near Proposition 3 to explicitly highlight this as an important and open problem for future research.
>
> [R1] Baik et al. Phase transition of the largest eigenvalue for nonnull complex sample covariance matrices. Annals of Probability, 2005
>
> [R2] Hansen et al. Deblurring Images: Matrices, Spectra, and Filtering, SIAM, 2006

---

> > ### Comment · Reviewer_m9NX · 2025-08-05
> >
> > I would like to thank the authors for their detailed and thoughtful response. It has clarified several aspects of the paper and improved my understanding of its contributions. I believe that the discussion above regarding the empirical evaluation of spectral alignment would make a valuable addition to the manuscript. I also appreciate the authors' clarification on the tightness of the upper bound. My concerns have been largely addressed, and I will adjust my score accordingly.

---

> > > ### Author Response · Authors · 2025-08-07
> > >
> > > Thank you for your feedback! We are glad to hear that our responses have addressed your concerns. We truly appreciate your time and engagement.

---

### Official Review · Reviewer_9cMm · 2025-07-03

**Clarity:** 3
**Significance:** 2
**Originality:** 3
**Rating:** 4
**Confidence:** 3

**Summary:**

- This paper studied the problem if using instrument variable for causal effect estimation when there is an existence of unobserved confounders. A potential unobserved confounder may affect both the treatment and effect, leading to incorrect causal effect estimation.
- In particular, the paper focused on the nonlinear two-stage least squares method for using instrument variables. Under the nonlinear 2SLS framework, regressions are done between the projected feature space of the treatment and instrument variable. Then the outcome is regressed   further against the predicted features.
- This paper investigated in the conditions where the nonlinear 2SLS approach perform well or badly, and identify three different regimes where the effectiveness of the framework is affected by the alignment between the structural function and the operator, and the rate of the singular value decay. Numerical simulations support the theoretical analysis.

**Questions:**

- how does the misalignment between h0 and the top eigenspaces of T compare to a fast eigenvalue decay? Which source is more significant in terms of damaging the effectiveness of the 2SLS?

**Ethical Concerns:**

["NO or VERY MINOR ethics concerns only"]

**Final Justification:**

I had concerns on the theoretical contribution and lack of more complex real-world data in the experimental secion. The authors response partially addressed these concerns, however my concern on the technical innovations remains. I will update my score to positive.

**Limitations:**

yes

**Quality:**

3

**Strengths And Weaknesses:**

Strength:
- Overall the paper is well-organized and presented clearly. This work considered an interesting angle of nonlinear 2SLS: what are the good or bad features to choose - which significantly affects the effectiveness of the regressions.  The framework was illustrated with practical examples, and each step of the analysis is sketched out in sequel.
- The result decomposes clearly the sources of errors in feature representation and approximations, demonstrate that the 2SLS with spectral features performs the best when h0 aligns with the top eigenspaces of T and its singular values decay slowly. This insight is useful for practitioners to perform the best practices when using 2SLS.

Weakness:
- although the paper studied nonlinear 2SLS and it's good / bad scenarios, the theoretical innovations are relatively weak. It is natural to see that due to the assumptions and design of the 2SLS framework, the best scenario occurs when h0 aligns with the top eigenspaces of T and its singular values decay slowly. This finding is more on the incremental side and lacks more innovations.
- the numerical experiments does not use any real-world datasets but only simulated data. The experimental results can be strengthened with more complex datasets and comparisons to other existing approaches.

---

> ### Author Rebuttal · Authors · 2025-07-30
>
> We thank the reviewer for the time spent in reviewing our manuscript, for the encouraging evaluation of our work, and for the helpful feedback. We address the questions and concerns of the reviewer below, and have proposed updates to the manuscript to address the latter.
>
> We would be grateful if the reviewer might consider increasing their score in case they are satisfied with the proposed changes.
>
> - **W1. Regarding the theoretical contribution**. While the high-level intuition that spectral alignment and singular value decay matter seems natural in retrospect, this understanding remained at the level of folklore, lacking a rigorous treatment. Our primary contribution is to systematize this intuition, characterizing the different performance regimes and, crucially, the failure modes of spectral feature learning for NPIV. By establishing a concrete "good-bad-ugly" taxonomy, along with a method to empirically evaluate which case a given problem falls into (see our answer to Weakness 1 of **Reviewer m9NX**), we offer valuable insights to both theorists and practitioners.
>
> - **W2. Regarding the use of real-world data**. We used synthetic data to cleanly isolate the effects of spectral alignment and singular value decay, as these ground-truth properties are unknown in real-world settings. We agree that validating our taxonomy on more complex datasets is an important step. As detailed in our response to Weakness 1 of **Reviewer m9NX**, we have a procedure to estimate spectral properties from data and have applied it to the dSprites dataset, confirming it falls into the "good" regime. We will add these results to the final manuscript.
>
> - **Q1. Comparison between misalignment and fast decay**. We split our answer in two parts: first, assuming we work with the true singular functions of $\mathcal{T}$ as features; second, when we work with learned features obtained from our constrastive learning procedure.
>
>     **Assuming one has access to the true normalised features** (i.e. the singular functions) $u_i,v_i$ of the conditional mean operator $\mathcal{T} =\sum_{i \geq 1 }\sigma_i v_i \otimes u_i$. The overall generalization error rate for $h_0$ depends on two key factors:
>
>     - *Operator ill-posedness*: The rate of singular value decay of $\mathcal{T}$, typically $\sigma_i \asymp i^{-p}$ for $p \geq 1$. A larger $p$ indicates faster decay and thus greater ill-posedness.
>
>
>     - *Solution smoothness/alignment*: The source condition
>     $$
>     \left\lVert h_0 \right\rVert^2_{r} := \left\lVert (\mathcal{T}^*\mathcal{T})^{-r/2}h_0 \right\rVert^2 = \sum_{i\geq 1} \frac{\langle h_0, u_i \rangle_{L_2(X)}^2}{\sigma_i^{2r}} < +\infty
>     $$
>     This condition, for $r \geq 0$, quantifies how "smooth" $h_0$ is relative to the spectral decomposition of $\mathcal{T}$, or equivalently, how quickly the alignment coefficients $\langle h_0, u_i \rangle_{L_2(X)}$ decay relative to the singular values. A larger $r$ implies $h_0$ is "smoother" or better aligned with the less ill-posed directions of $\mathcal{T}$. Note that under the "bad" scenario where the eigendecay is fast ($p$ is large), a stronger alignment is necessary for $\left\lVert h_0 \right\rVert^2_{r}$ to be finite for a large value of $r$. While in the "good" scenario with $p \sim 1$, $\left\lVert h_0 \right\rVert^2_{r} < +\infty$ is more likely to hold with a large value of $r$.
>
>     Under the source condition, the approximation error when using as features the top-$d$ singular functions of $\mathcal{T}$ is:
>     $$
>     \sum_{i>d} \langle h_0, u_i \rangle_{L_2(X)}^2 \leq \sigma_d^{2r} \sum_{i>d} \frac{\langle h_0, u_i \rangle_{L_2(X)}^2}{\sigma_i^{2r}} \leq \sigma_d^{2r}\left\lVert h_0\right\rVert^2_{r}.
>     $$
>     Combined with the eigenvalue decay (involving $p$), we obtain from Corollary 1 in our submission that:
>     $$
>     \left\lVert \hat{h}-h_0\right\rVert_{L^2}=O_p\left(\left\lVert h_0 \right\rVert_{r}\sigma_d^{r}+\sqrt{\frac{d}{n\sigma_d^2}}\right) = O_p\left(\frac{\left\lVert h_0 \right\rVert_{r}}{d^{r\times p}}+d^{1/2 +p}\sqrt{\frac{1}{n}}\right).
>     $$
>     Balancing both terms in $d$ as a function of $n$, we obtain
>     $$
>     \left\lVert\hat{h}-h_0 \right\rVert_{L^2}=O_p\left(n^{-\frac{r}{2(r+1) + p^{-1}}}\right).
>     $$
>     with $d(n) = n^{\frac{1}{1 + 2p(r+1)}}$. This bound is tight as a matching lower bound is obtained in [1, Theorem 1-(i)].
>
>     From this tight rate, we see that both the ill-posedness ($p$) and the relative smoothness ($r$) fundamentally damage the effectiveness of 2SLS, and their impacts are inherently intertwined.
>
>     - A fast decay (large $p$) directly slows down the rate regardless of $r$. It means singular values drop quickly, making it fundamentally harder to resolve higher-frequency components of $h_0$.
>     - A low relative smoothness (small $r$) means $h_0$ has significant components aligned with directions corresponding to very small singular values, even after accounting for the singular value decay. This also directly slows the rate, effectively rendering the problem very hard or even impossible ($r \to 0_+$, makes the rate vacuous, representing our "ugly" scenario).
>
>     **When working with learned features via contrastive learning.** We see from our Proposition 3 that when we learn the spectral features using contrastive learning, we incur an additional penalty $\varepsilon/\sigma_d$ in the upper bound, where $\varepsilon$ is the optimality gap on the contrastive learning loss. The alignment of $h_0$ does not enter this factor, as learning the eigensubspaces of $\mathcal{T}$ is unrelated to $h_0$. This term shows that a faster eigendecay makes the weight of the error incurred by contrastive learning grow faster. Intuitively, features corresponding to small singular values can be empirically difficult to learn, even with a very large number of $(Z,X)$ samples.
>
>     We will add this discussion as a remark in the main paper, with full details in the Appendix. We think that it is a great addition to the discussion in our manuscript.
>
> [1] Xiaohong Chen, Markus Reiss. "On rate optimality for ill-posed inverse problems in econometrics." Econometric Theory, Vol. 27, No. 3, INVERSE PROBLEMS IN ECONOMETRICS (June 2011), pp. 497-521.

---

> > ### Comment · Reviewer_9cMm · 2025-08-09
> > **response to author rebuttal**
> >
> > I appreciate the authors for the detailed response in the rebuttal and the additional results with regard to my concern on real-world dataset results. I find all my points addressed and I will update my score accordingly.

---

### Decision · Program_Chairs · 2025-09-17

**Decision:**

Accept (poster)

**Comment:**

This work analyzes nonparametric IV regression with spectral features, deriving generalization bounds and showing that performance depends on spectral alignment and eigenvalue decay, leading to a “good–bad–ugly” taxonomy that is theoretically characterized and empirically validated.

All reviewers viewed this paper positively and recommend acceptance. In my own reading this work makes a good contribution worthy of publication at NeurIPS.